# Structure and electromechanical coupling of a voltage-gated Na⁺/H⁺ exchanger

Hyunku Yeo[1,2], Ved Mehta[1,2], Ashutosh Gulati[1,2] & David Drew[1✉]

Voltage-sensing domains control the activation of voltage-gated ion channels, with a few exceptions[1]. One such exception is the sperm-specific Na⁺/H⁺ exchanger SLC9C1, which is the only known transporter to be regulated by voltage-sensing domains[2–5]. After hyperpolarization of sperm flagella, SLC9C1 becomes active, causing pH alkalinization and CatSper Ca²⁺ channel activation, which drives chemotaxis[2,6]. SLC9C1 activation is further regulated by cAMP[2,7], which is produced by soluble adenyl cyclase (sAC). SLC9C1 is therefore an essential component of the pH–sAC–cAMP signalling pathway in metazoa[8,9], required for sperm motility and fertilization[4]. Despite its importance, the molecular basis of SLC9C1 voltage activation is unclear. Here we report cryo-electron microscopy (cryo-EM) structures of sea urchin SLC9C1 in detergent and nanodiscs. We show that the voltage-sensing domains are positioned in an unusual configuration, sandwiching each side of the SLC9C1 homodimer. The S4 segment is very long, 90 Å in length, and connects the voltage-sensing domains to the cytoplasmic cyclic-nucleotide-binding domains. The S4 segment is in the up configuration—the inactive state of SLC9C1. Consistently, although a negatively charged cavity is accessible for Na⁺ to bind to the ion-transporting domains of SLC9C1, an intracellular helix connected to S4 restricts their movement. On the basis of the differences in the cryo-EM structure of SLC9C1 in the presence of cAMP, we propose that, upon hyperpolarization, the S4 segment moves down, removing this constriction and enabling Na⁺/H⁺ exchange.

Directed sperm motility is essential for fertilization[10]. From corals to human, metazoan sperm mobility is directed by a highly conserved signalling cascade in which sperm motility is stimulated by the pH-dependent activation of the cAMP-producing enzyme sAC[8,9]. In response to various external stimuli, hyperpolarization of the flagellum membrane occurs, which in turn activates the sperm-specific Na⁺/H⁺ exchanger (NHE) SLC9C1[4,8], causing an increase in the intracellular pH[2,4,8]. Both pH alkalinization and increased cAMP production by sAC activate the CatSper Ca²⁺ channel[6], which drives chemotaxis[8,9] (Fig. 1a). Key to sperm signalling and motility are the changes in intracellular pH elicited by activation of SLC9C1[4]. Homozygous *SLC9C1* mutations in male humans have been reported to cause poor sperm motility and infertility[11]. A mouse *Slc9c1* knockout renders sperm immotile and male *Slc9c1*-knockout mice are infertile[4,7], although bulk pH itself is unaffected[4]. Rather than pH homeostasis, it seems that SLC9C1 has evolved as a regulatory switch for the pH-dependent activation of sAC and subsequent cAMP signalling, which are essential for both sperm motility and maturation in the female reproductive tract[12,13].

SLC9C1 belongs to the NHE family of ion transporters[14], which facilitate the electroneutral exchange of cations (Na⁺/Li⁺/K⁺) and protons (H⁺) across membranes to regulate intracellular pH, sodium levels and cell volume[14]. In mammals, the 13 different NHE isoforms are spread across the three distinct clades: SLC9A (NHE1–9), SLC9B (NHA1–2)

and SLC9C1–2[14]. SLC9C1 and all other NHEs are thought to be physiological homodimers[14,15], with each monomer made up of a transporter unit and a C-terminal regulatory domain of varying length[14]. The transporter module is made up of the NhaA-fold, named after the first crystal structure of NhaA from *Escherichia coli*[16]. Cryo-EM structures of SLC9A and SLC9B members[17–19] have shown that they are most similar to bacterial Na⁺/H⁺ antiporters with 13 transmembrane (TM) segments[20–22]. The transporter module consists of two distinct domains, a dimerization domain and an ion-transporting core domain. The core domain undergoes global, elevator-like structural transitions to translocate ions across the membrane against the anchored dimerization domain[19,23,24]. The C-terminal regulatory domain has large sequence variation between SLC9A isoforms[14] and can interact with a multitude of different proteins, such as Ca²⁺/calmodulin[25], which allosterically regulates NHE activity in an isoform-specific manner[14].

SLC9C1 has a very different C-terminal regulatory domain compared with all of the SLC9A members[14,18,19]. Rather than containing a partially disordered cytoplasmic tail[26], it contains four additional TM helices that form an S1–S4 voltage-sensing domain (VSD), and a distal cyclic-nucleotide-binding domain (CNBD)[2,4] (Fig. 1b). The S4 VSD renders SLC9C1 a transporter that is activated only in response to hyperpolarization (−95 mV), and the CNBD bind cAMP to enable SLC9C1 to become activated closer to the resting membrane potential

[1]Department of Biochemistry and Biophysics, Science for Life Laboratory, Stockholm University, Stockholm, Sweden. [2]These authors contributed equally: Hyunku Yeo, Ved Mehta, Ashutosh Gulati. ✉e-mail: ddrew@dbb.su.se

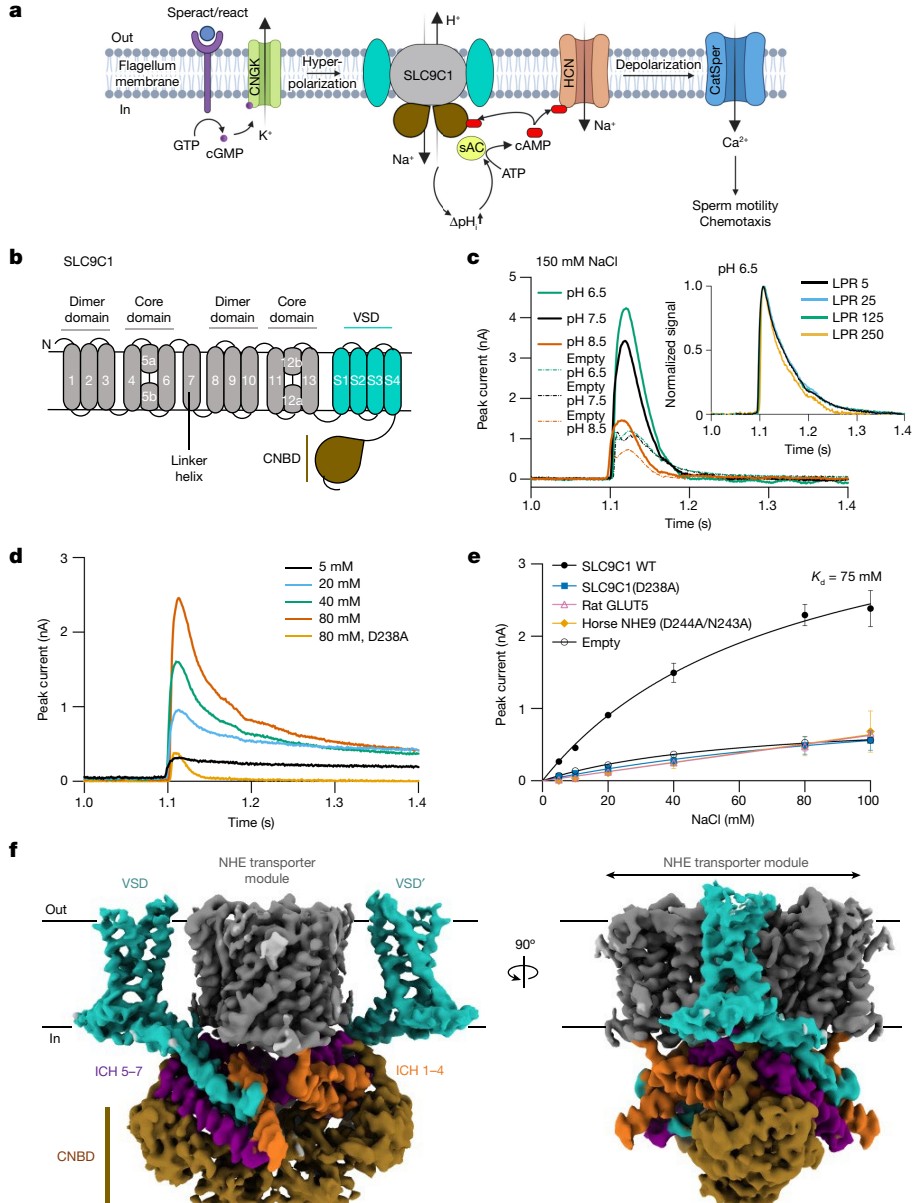

**Fig. 1 | Functional characterization and structure determination of sea urchin SLC9C1. a**, SLC9C1 is essential for the pH-mediated regulation of sAC, which leads to CatSper Ca²⁺ channel activation and thereby drives sperm motility. In sea urchin sperm, the binding of the chemoattractant speract to its extracellular receptor enhances intracellular cGMP production, CNGK channel opening and the hyperpolarization-induced activation of SLC9C1[2,3,5,8]. The diagram was created using BioRender.com. **b**, The NHE transporter module has 13 TM segments, forming a core ion-transport domain and dimer domain (grey). Downstream of the NHE transporter module is an S4 VSD (cyan) and a CNBD (brown). **c**, SSM-based electrophysiology analysis. Transient currents after the addition of 150 mM NaCl were recorded from SLC9C1 proteoliposomes and empty liposomes at a symmetrical pH of 6.5, 7.5 and 8.5 as labelled. Inset: normalized peak currents in response to 80 mM NaCl at pH 6.5 for varied LPR. **d**, Representative transient currents at pH 6.5 with varying concentrations of

NaCl. The key SLC9C1 ion-binding aspartate D238A mutant response to 80 mM NaCl is also shown. **e**, Fit of the transient currents as a function of Na⁺ concentrations for SLC9C1(WT), SLC9C1(D238A), rat fructose transporter GLUT5, horse NHE9(D244A/N243A) non-functional variant and empty liposomes at a symmetrical pH of 6.5 as labelled. Data are mean ± s.d. of $n = 3$ titrations (sensors) for SLC9C1(WT), SLC9C1(D238A), empty liposomes and a range of $n = 2$ titrations for rat GLUT5 and horse NHE9. The $K_d$ for Na⁺ is shown for WT SLC9C1. **f**, Cryo-EM maps of the sea urchin SLC9C1 homodimer in GDN detergent ($C_2$ symmetry). The NHE transporter module (grey), the VSD domain (cyan) and CNBD domain (brown) are shown. Connecting the NHE transporter to the VSD are intracellular helices ICH1 to ICH4 (orange), and connecting the VSD domain to the CNBD are intracellular helices ICH5 to ICH7 (purple). **a**, Adapted from ref. 2, CC BY 4.0.

of sperm (−50 mV)[2] VSDs are thought to have evolved independently from ion channels, and there are a few examples of non-channel VSD domains coupled to soluble enzymes[1]. However, the coupling of a S4 VSD to a transporter appears to be unique to SLC9C1. Furthermore, electrophysiological experiments have demonstrated that the gating currents ($g_q$) of SLC9C1 correspond to 3.1 elemental charges ($e_0$) (ref. 2)

as seen in bona fide ion channels, for example, K⁺ Shaker channels[27], rather than non-channel VSD proteins, which have lower $g_q$ values of 1–1.6 $e_0$ (ref. 28). Here we set out to determine the architecture of this unique voltage-gated transporter and to establish how ion exchange can be allosterically regulated by S4 VSDs that are typically tethered to voltage-gated ion channels.

## SLC9C1 isolation and electrophysiology

Sea urchins have long been used to develop models of fertilization[8,29], and the voltage-dependent NHE SLC9C1 was therefore first described in sea urchin spermatozoa[3,5] before its characterization in mammals[4]. To date, owing to technical difficulties in recombinant expression of the mammalian counterparts, SLC9C1 from the sea urchin *Strongylocentrotus purpuratus*, which shares around 30% sequence identity with human SLC9C1, is the best understood model system[2] (Supplementary Fig. 1). Heterologous expression of *S. purpuratus* SLC9C1 in mammalian cells yields a functional transporter that has a voltage of half-maximal activation ($V_{1/2}$) of −95 mV, which shifts to −75 mV in the presence of 1 mM cAMP[2]. After extensive optimization, we succeeded in purifying microgram amounts of detergent-stable full-length *S. purpuratus* SLC9C1 after its recombinant expression in large-scale cultures of HEK293 cells (Methods, Extended Data Fig. 1a and Supplementary Fig. 2). Purified SLC9C1 was reconstituted into liposomes made from yeast polar lipids for solid-supported membrane (SSM)-based electrophysiology recordings (Supplementary Fig. 3a), which is a more sensitive technique than patch-clamp electrophysiology for low-turnover transporters[30]. In this technique, proteoliposomes are adsorbed onto an SSM, and the charge translocation of ions is measured by capacitive coupling of the supporting membrane[30]; this method can also detect pre-steady-state currents of the half-reaction in electroneutral transporters[31].

Peak currents for SLC9C1 proteoliposomes were clearly measurable after the addition of 150 mM NaCl at pH 6.5, 7.5 and 8.5 (Fig. 1c). In contrast to bacterial NHEs and SLC9B members, which that display higher activity at pH 7.5 than at pH 6.5 (refs. 17,32,33), the highest activity for SLC9C1 was observed at pH 6.5, matching the acidic pH 6.7 cytoplasm of sea urchin sperm flagella under similar conditions[5]. Given that SLC9C1 is inactive at 0 mV, we assume that the majority of the transient current signal represents local ion-binding events, rather than bulk transport, similar to what has been observed for non-transported substrates, which can give substantial peak currents in SSM[34]. Consistent with this interpretation, decreasing the lipid-to-protein ratio (LPR) produced no systematic differences in the calculated time constant from decay currents (Fig. 1c and Supplementary Fig. 3b,c)—a transported charge should increase current decay towards lower LPRs[34]. Increasing Na[+] concentrations at pH 6.5 confirmed that the peak currents were significantly higher than the signal observed for either empty liposomes or a purified SLC9C1 mutant in which the ion-binding aspartate Asp238 was substituted with alanine (Fig. 1d and Supplementary Figs. 3a and 4). The peak currents from either a non-functional ion-binding aspartate variant of horse NHE9[18] or the unrelated rat fructose transporter GLUT5 were also of similar amplitude to the SLC9C1(D238A) variant (Fig. 1e and Supplementary Fig. 4). The apparent binding affinity of sea urchin SLC9C1 for Na[+] was $K_d$ = 75 mM (Fig. 1e), which is lower than the Michaelis constant ($K_M$) estimates of mammalian NHEs (20 to 40 mM)[14], but understandable given the high salinity of sea water.

## Cryo-EM structures of sea urchin SLC9C1

SLC9C1 samples either purified in the detergent GDN or further reconstituted into MSP1E2 nanodiscs were optimized for grid preparation and cryo-EM data acquisition (Methods). Cryo-EM maps for SLC9C1 in detergent and nanodiscs could be reconstructed at resolutions of 3.2 Å and 3.0 Å, respectively (Extended Data Table 1, Extended Data Figs. 1 and 2 and Supplementary Fig. 2). The cryo-EM maps after applying $C_2$ symmetry were well resolved for the transporter module and the CNBD domains (Fig. 1f), but the map densities for the VSDs were poorer in both detergent and nanodisc 3D reconstructions. We subsequently applied symmetry expansion and masked refinement, which significantly improved the map density of the VSDs in the GDN sample,

such that most side chains could now be modelled at a resolution of 3.0 to 3.5 Å (Extended Data Figs. 1 and 3 and Supplementary Video 1). The maps obtained in nanodiscs showed poor map density for the VSDs (Extended Data Fig. 2 and Supplementary Fig. 5a,b). With a subset of particles processed in $C_1$ symmetry, the map density of the VSD in nanodiscs was improved to confirm the same overall positioning of the VSD as in detergent, but with no clear side-chain map density (Extended Data Fig. 2 and Supplementary Fig. 5c). The cryo-EM maps for the transporter module were at a higher resolution in some regions of the nanodisc reconstruction, with a local resolution of up to 2.5 Å in most regions (Extended Data Figs. 3 and 4). Overall, apart from a few flexible loops and the most distal C-terminal residues, we could confidently build a model of the entire SLC9C1 structure (Supplementary Fig. 1 and Extended Data Figs. 3 and 4).

## The NHE transporter module of SLC9C1

The transporter module of the SLC9C1 monomer consists of 13 TM segments, and is overall similar to the mammalian NHE1[19], NHE3[35] and NHE9[18] structures (Fig. 2a and Supplementary Fig. 6a). The SLC9C1 dimerization interface is formed predominantly by tight interactions between TM1 on one monomer and TM8′ on the other, burying a total interface area of around 1,900 Å[2] in the detergent structure (Fig. 2a and Supplementary Fig. 6b). As observed in other Na[+]/H[+] exchanger structures[18,19], there is a hydrophobic cavity between protomers (Fig. 2b and Extended Data Fig. 5a). In the detergent SLC9C1 structure, cylindrical density indicated the presence of two fatty-acid tails (Fig. 2b and Extended Data Figs. 3 and 5a), as seen in the structure of human NHE1[19]. In the nanodisc structure, cryo-EM map density at a local resolution of 2.5 Å supported the modelling of two phosphatidic acid (PA) lipids at the dimerization interface (Fig. 2c and Extended Data Figs. 2b, 4 and 5a,b). To support this structural observation, SLC9C1 was repurified in DDM + CHS (DDM/CHS), instead of the more stabilizing GDN detergent (Supplementary Fig. 6c). In DDM/CHS, the SLC9C1 protein migrated as both a dimer and monomer (Extended Data Fig. 5c). The purified SLC9C1 protein was further heated at 45 °C for 10 min in the presence of PA, PC, PS or PE lipid, yet only the addition of PA improved the resistance of SLC9C1 to heat denaturation (Extended Data Fig. 5c and Supplementary Fig. 6c).

The presence of PA lipids probably stabilized a more compacted SLC9C1 homodimer in the nanodisc structure, which was evident by the repositioning of helices at the dimerization interface (Fig. 2b–d and Extended Data Fig. 5a,b). In particular, the N-terminal tail contains two histidine residues, His71 and His73, that, in the detergent structure, are interacting with Asp134 located between TM2′–TM3′ of the neighbouring protomer (Extended Data Fig. 5b). However, in the presence of PA, His73 interacts with the lipid, and the TM2′–TM3′ extracellular region has rearranged to coordinate the PA headgroup through water-bridged His132 and Asp129. The lipid-induced rearrangement observed in SLC9C1 is centred around an extracellular helical motif (ECH1) that is not present in the NHE1[19], NHE3[35] and NHE9[18] structures (Fig. 2a and Supplementary Fig. 6a). Given that specific lipids can modulate the oligomerization and activity of homologous NHEs[17,18], it seems probable that the lipid PA fills an allosteric role in SLC9C1, possibly linked to extracellular pH regulation. In addition to these lipids, peripheral lipids matching PA and PC were further modelled next to the flexible TM7 linker in the nanodisc and detergent structures, respectively (Extended Data Figs. 3 and 4 and Supplementary Fig. 5a).

The structure of SLC9C1 shares the highest structural similarity to human SLC9A1 (NHE1) and horse SLC9A9 (NHE9) that were captured in an inward-facing conformation[18,19] (Supplementary Fig. 6a). Consistent with an inward-facing structure for SLC9C1, a highly negatively charged funnel extends from the ion-binding site to the cytoplasmic surface (Fig. 2b,e). Near to the base of the cavity, the highly conserved aspartate residue Asp238 (TM6) is located (Fig. 2e) that, in related Na[+]/H[+]

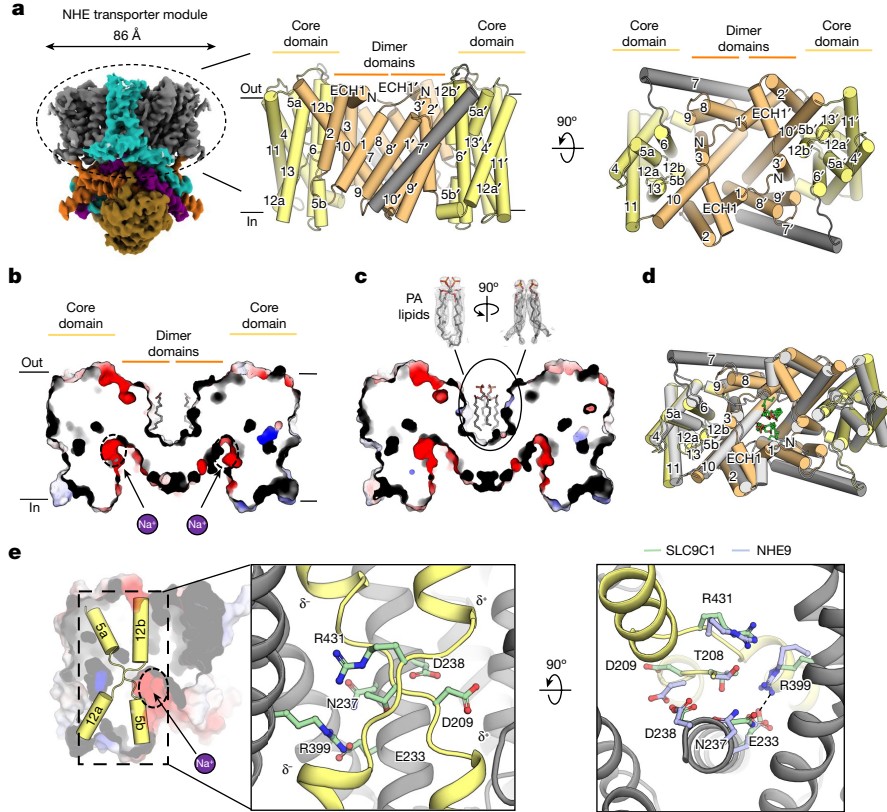

**Fig. 2 | The structure of the NHE transporter module and PA-mediated remodelling. a**, Cryo-EM map of SLC9C1 in GDN detergent with the NHE transporter module circled (left). Middle, cartoon representation of the NHE transporter homodimer, with the dimer domain comprising TM1–TM3 and TM8–TM10 (light orange), and the core domain comprising TM4–TM6 and TM11–TM13 (light yellow). The core and dimer domains are connected by the linker helix TM7 (grey). Right, as described for the middle image, but shown from the extracellular side. **b**, Cross-section of the electrostatic surface along the side view of the SLC9C1 homodimer in GDN detergent showing the negatively charged inward-facing cavity, the location of the ion-binding site (dotted black circle and purple Na⁺ ion) and the extracellular, hydrophobic gap between protomers with two lipid tails bound (grey sticks). **c**, As in **b**, for the SLC9C1 homodimer in nanodiscs, showing the compaction of the extracellular gap and the binding of two PA lipids (grey sticks) and the cryo-EM map density (grey mesh). **d**, SLC9C1 homodimer in detergent (light orange, dimer domain; yellow,

core domain) superimposed with the SLC9C1 homodimer in nanodiscs (grey), showing the compaction of the dimerization interface in the presence of PA lipids (green sticks). **e**, Electrostatic surface representation of the SLC9C1 monomer and cartoon representation of the half helices TM5a-b and TM12a-b (yellow) and ion-binding site (dotted black circle and purple Na⁺ ion): the crossover half helices are unique to the NhaA-fold and the half-helical dipoles that are formed are denoted (left). Middle, in SLC9C1 Arg431 (green-stick) neutralizes the negatively charged half-helical dipole, and Asp209 (green-stick) neutralizes the positively charged half-helical dipoles. Asn237 and Asp238 in TM6 form the canonical ND motif[15,37], characteristic of electroneutral NHEs. Right, ion-binding site residues for sea urchin SLC9C1 (green sticks) and the corresponding position of ion-binding site residues from horse NHE9 (Protein Data Bank (PDB) 6Z3Z; light blue). In SLC9C1 a salt-bridge (dashed line) is formed between Arg399 in TM11 and Glu233 in TM6.

exchangers, is essential for ion binding and transport[14,20,22,23,36]. The core domain contains two discontinuous helices, TM5a-b and TM12a-b, that cross over each other near the centre of the membrane. The residue Asp209 neutralizes the two positively charged half-helical dipoles and Arg431 neutralizes the negatively charged dipoles (Fig. 2e). The side chains of Asn237 and Asp238, which make up the well-conserved ion-binding ND motif of the electroneutral NHEs[15,37], are positioned similarly to in other NHE structures (Fig. 2e). The additional salt bridge between Asp233 and Arg399 (TM11), stabilizing the ion-binding site in the core domain in other NHE members[18,19], is likewise conserved (Fig. 2e and Supplementary Fig. 1). Consistently, substitution of Arg399 to alanine abolished Na⁺/H⁺ exchange activity in CHO cells expressing sea urchin SLC9C1[2]. On the basis of the recent K⁺/H⁺ exchanger structure in complex with K⁺ (ref. 38) and MD simulations of Na⁺/H⁺ exchangers[17,18,23,33], Na⁺ is thought to be coordinated entirely in the core domain between the cross-over helices and Asn237 and Asp238 residues. Overall, the structure of the transporter module and ion-binding site is entirely consistent with that of a typical mammalian NHE, with most structural differences located at the oligomerization interface (Fig. 2e and Extended Data Fig. 5a).

## VSDs of SLC9C1

In both the detergent and nanodisc structures, along the membrane plane and viewed from the extracellular surface, the VSDs are located perpendicular from the dimerization interface and on either side of the SLC9C1 transporter module (Fig. 3a and Supplementary Fig. 5). The VSDs are indirectly connected to the SLC9C1 transporter module by a number of intracellular helices (Fig. 3a,b). The first intracellular helix (ICH1), starts some six residues from the end of TM13, and extends at a 45° angle to the opposite-facing VSD′ before looping back to the transporter module through a short helix (ICH2) and a flexible loop, which we were unable to model owing to poor map density (Fig. 3a). The flexible loop is continued by an antiparallel helix–turn–helix motif (ICH3 and ICH4) positioned below its own transporter subunit, and then a flexible S1 linker, which extends at 45° to the beginning of the oppositely located VSD (Fig. 3a). Thus, the domain-swapped ICH1–ICH2, together with ICH3–ICH4, indirectly connect the VSDs to one other. However, there are no direct interactions from the VSDs to the transporter module itself; instead, they are held approximately 20 Å apart (Fig. 3a). On the basis of the SLC9C1 structures presented here,

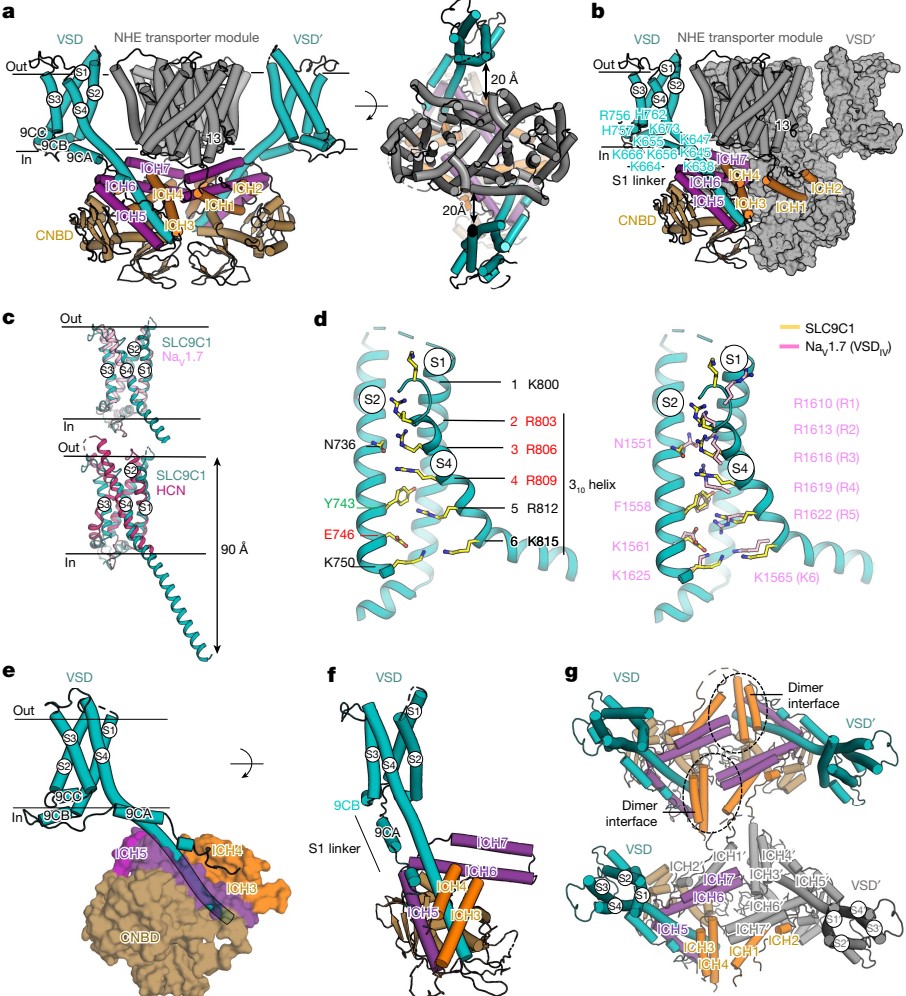

**Fig. 3 | The VSD domain in SLC9C1 is most similar to VSD_IV from human Na_V1.7. a**, The SLC9C1 homodimer (side view), with the NHE transporter (grey), VSD domains (cyan), CNBD (brown), and the intracellular helices ICH1–ICH4 (orange) and ICH5–ICH7 (purple), with one protomer labelled (left). Right, as described for the left image, but the extracellular view with the distance between the NHE and VSD domains labelled. **b**, Cartoon of one protomer as in **a**, and the other protomer as a surface representation (grey) highlighting the domain-swapped ICH1–ICH2 helices. **c**, Superimposition of the SLC9C1 VSD (cyan) with VSD_IV from Na_V1.7 (PDB: 7XMG; light pink, root mean squared deviation (r.m.s.d.) of 2.4 Å) and with the distal end of S4 omitted for clarity (top), and with VSD from the HCN channel (PDB: 5U6P; dark pink, r.m.s.d of 5.4 Å), shown as above, but with the entire S4 helix (bottom). **d**, VSD of SLC9C1 with S3 omitted to highlight key residues (yellow sticks) (left). The occlusion-forming Tyr743 in S2 and the negatively charged residue Glu746, together with

Asp767, make up the conserved charge transfer centre[43]. S4 is in the up-state and gating-charge residues Arg803 (R2) to Lys815 (K6) extend from the 3_10 helix. The gating-charge residues Arg803 (R2), Arg806 (R3) and Arg809 (R4) are positioned above the charge transfer centre, and probably account for the three measured gating-charges[2]. Right, as described for the left image, with key residues of SLC9C1 (yellow sticks) and VSD_IV of human Na_V1.7 (pink sticks) labelled. **e**, The VSD from SLC9C1 (cyan) and a semi-transparent surface representation of ICH3–ICH4 (orange) and ICH5–ICH6 (purple) form extensive contacts to extramembranous S4 and the CNBD (brown). **f**, A 45° rotation of the VSD shown in **e** (all cartoon representation) and the S1 linker with the 9CA and 9CB helices are also highlighted. **g**, Cartoon of the VSD and ICH network (coloured as described in **a**) as viewed from the extracellular side with the NHE transporter module omitted (top). Bottom, as described for the top image, but one protomer is rendered in grey.

one might expect that the VSDs are able to operate independently from the NHE transporter module. Indeed, in sea urchin SLC9C1, the Arg399Ala variant abolishing transporter activity did not perturb gating currents in CHO cells[2]. Thus, although the VSDs in SLC9C1 can probably operate independently, the transporter is unable to work unless the VSDs move in response to hyperpolarization.

The VSDs in SLC9C1 have the classical VSD architecture as seen in K⁺-shaker K_V channels[39] (Extended Data Fig. 6a,b). Using Foldseek[40], the VSDs in SLC9C1 were surprisingly most similar to VSD_IV from the human voltage-gated Na⁺ channel structure of Na_V1.7 in the active up-state[41,42] (Fig. 3c). In particular, the S4 segment of SLC9C1 contains six gating-charge residues—Lys800 (K1), Arg803 (R2), Arg806 (R3), Arg809 (R4), Arg812 (R5) and Lys815 (K6)—and is similar in both

sequence and structure to the VSD_IV of Na_V channels[41,42] (Fig. 3c,d and Extended Data Fig. 6a). The occlusion-forming aromatic Tyr743 in S2 and the negatively charged residues Glu746 and Asp767, salt bridged with Arg812 (R5), further make up the conserved charge transfer centre[43] (Fig. 3d). Gating-charge residues Arg803 (R2) to Lys815 (K6) are located on the expected 3_10 helix of S4, with Arg803 (R2) forming a salt bridge to Glu698 (on S1), Arg806 (R3) salt bridged to Glu698 (in S1) and hydrogen bonded to Asn736 (in S2), and Arg809 (R4) hydrogen bonded to Asn691 (on S1) (Fig. 3d and Extended Data Fig. 6b,c). The majority of the gating-charge residues in the VSD domain are conserved and similarly positioned between SLC9C1 and VSD_IV of Na_V1.7[41,42] (Fig. 3d). One notable difference is an additional aspartate Asp777 in S3, which is not present in Na_V1.7[41,42] and forms an interaction with the backbone of

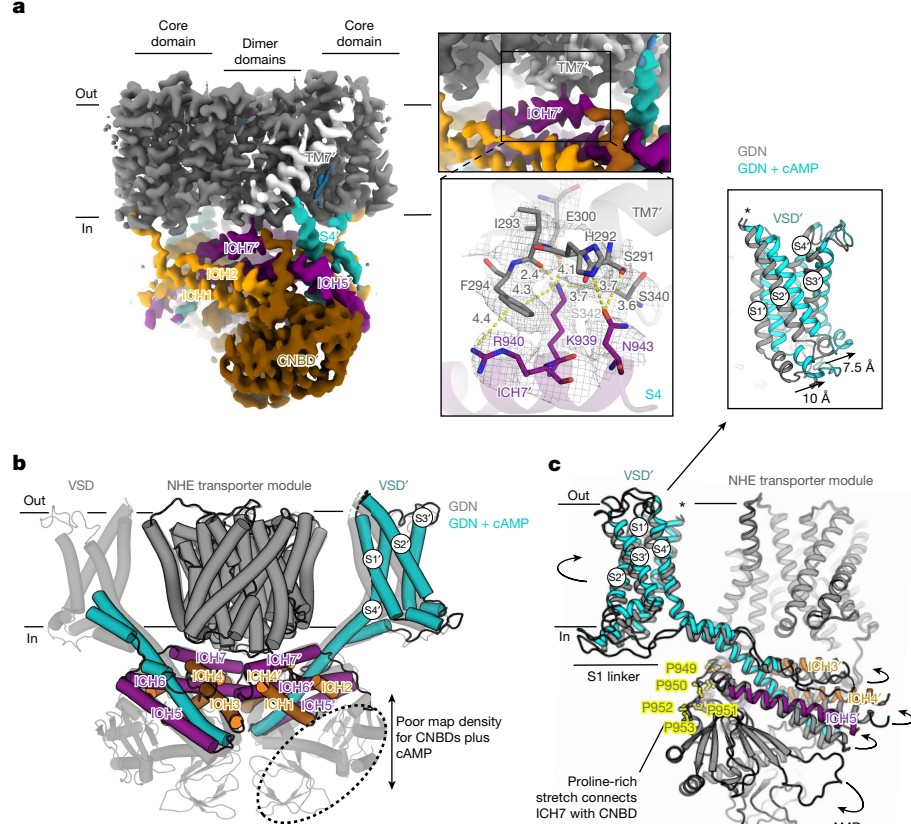

**Fig. 4 | Autoinhibition by ICH7 and cAMP binding primes SLC9C1 for activation. a**, Cryo-EM map of SLC9C1 in nanodiscs (coloured as in Fig. 3a) highlighting the only interaction between the cytoplasmic assembly (ICH7, purple) and the transporter module (TM7, light grey) (left). A peripherally located PA lipid next to TM7′ is also shown (blue). Top right, magnified view of the maps showing the interactions between Lys939 in ICH7′ and TM7′. Bottom right, cartoon representation of SLC9C1 with a cryo-EM mesh (grey) depicting the detailed interactions between ICH7′ (purple) and TM7′ (grey) with labelled residues (stick form). The yellow dashed lines show mostly hydrogen-bond distances of 2.4 to 4.0 Å between Lys939 and the backbone of TM7′ and the side chains Ser340 and Ser342. Further ICH7′–TM7′ interactions are formed between Asn943 and His292 and a π-cation interaction between Phe294 and

Arg940. **b**, The SLC9C1 detergent structure with cAMP addition (coloured as in **a**) superimposed onto the apo SLC9C1 detergent structure (transparent grey). The cryo-EM map density was mostly absent for the CNBD with cAMP addition, as shown in Extended Data Fig. 9b, except for poor map features for one of the CNBDs (dotted ellipsoid). **c**, Superimposition of SLC9C1 structures in detergent without cAMP (grey) and with cAMP (coloured as in **a**). The binding of cAMP to the CNBD probably detaches the domain from S4 and from ICH5, which propagates with the displacement of ICH3, ICH4 and ICH5 as also shown in Extended Data Fig. 10b. The change in the position of the ICHs has altered the positioning of the VSD. Inset: magnified view showing the anti-clockwise movement of the S1–S3 helices when comparing the SLC9C1 structures obtained with (coloured) and without (grey) cAMP addition.

Arg803 (Extended Data Fig. 6c). However, despite subtle differences, the overall structural similarity to Na$_v$1.7 is highly similar and, because 3.1 $e_0$ has been measured by electrophysiology analysis of SLC9C1[2], the gating currents almost certainly correspond to the movement of the arginine gating-charge residues R2 to R4, which are positioned above the charge transfer centre. Consistently, it has been shown that replacing Arg803 (R2) with glutamine in sea urchin SLC9C1 shifts the $V_{1/2}$ by −24 mV to −118 mV (ref. 2).

A fundamental difference between voltage-gated SLC9C1 and voltage-gated ion-channels is their electromechanical coupling. In SLC9C1, there is no S4–S5 linker, which, in ion channels, physically connects the voltage sensor to the pore domain[44]. Rather, the S4 helix in SLC9C1 continues some 60 Å in length, and is considerably longer than even the very long S4 helix observed in hyperpolarization-activated cyclic nucleotide-gated (HCN) channels[45] (Fig. 3c). S4 appears to be fairly rigid at 0 mV, based on the fact that the cryo-EM map density for S4 is at a higher resolution than the rest of the VSD in both the nanodisc and detergent structures (Extended Data Fig. 3, Supplementary Fig. 5 and Supplementary Video 1). Notably, the extramembranous S4 helix tunnels between two antiparallel helix–turn–helix motifs ICH3–ICH4 and ICH5–ICH6 (Fig. 3e,f) and forms extensive ICH interactions, burying

a total surface area of 2,100 Å$^2$ (Extended Data Fig. 7). The helix–turn–helix ICH3–ICH4 is connected to the start of S1 through a long flexible linker some 70 Å in length, whereas ICH5–ICH6 is connected directly to the end of S4 helix through a short loop of four residues (Fig. 3e,f). The partially ordered S1 linker contains two helical stretches labelled 9CA and 9CB, which connect the end of ICH4 with the beginning of S1. Eight positively charged residues in total are located along the 9CA and 9CB helical stretch (Fig. 3b,e,f). The positively charged interface, together with the flexible S1 linker, should enable the mobile S4 element to move against the S1–S3 helices. Taken together, as the ICHs are connected only to the S4 helix (Fig. 3e–g), upon hyperpolarization, movement of S4 to the down-state as in Na$_v$ channels[46–48] should also induce downward movement of the ICH3–ICH4 and ICH5–ICH6 helices, which are clamped extensively onto the S4 helix.

## cAMP priming and SLC9C1 activation

The CNBD in SLC9C1 is similar to the CNBD seen in HCN channels[45] (Fig. 4a and Extended Data Fig. 8a,b). However, the CNBD has an additional downstream helix (αD) and β-roll domain with three positively charged residues that interact with their equivalents from the second

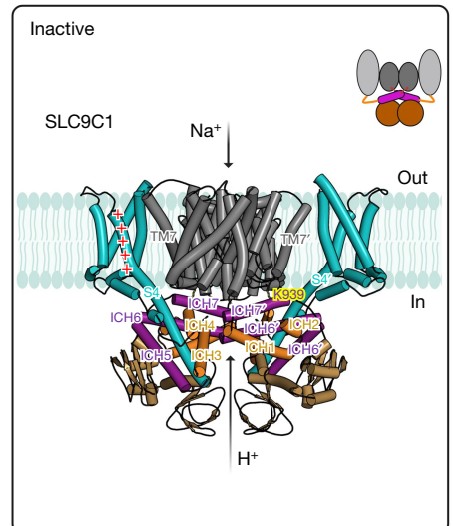
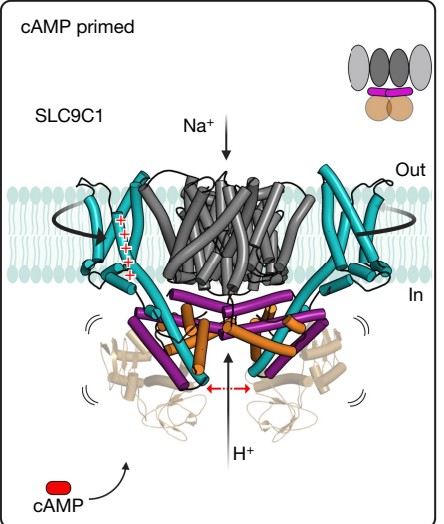
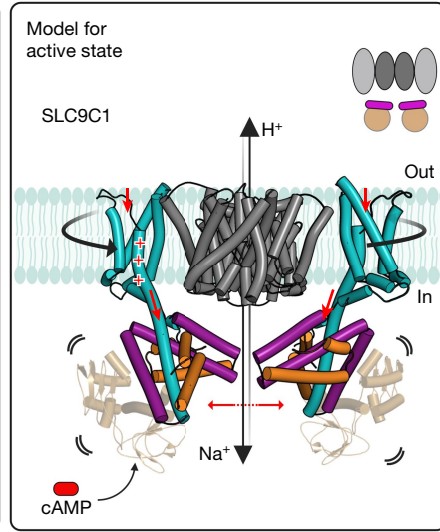

**Fig. 5 | Schematic of the voltage-dependent activation of sperm-specific SLC9C1.** At 0 mV, the SLC9C1 transporter (grey) is able to bind to Na[+], as shown here by SSM-based electrophysiology and as evident from the open inward-facing cavity, but is inactive because the very long S4 in the VSD (cyan) is in the up-state, which pulls up the clamped ICH3–ICH4 (orange) and ICH5–ICH6 (purple) helices so that ICH7 physically restricts the transporter domains from moving (left). In particular, Lys939 (yellow) in ICH7 is making a number of polar interactions to the linker helix TM7, which needs to rotate during an elevator alternating-access mechanism[23,24]. In the inactive state, the CNBDs (brown) further contribute to homodimerization and inactivation by stabilization of the cytosolic regulatory network. Middle, the addition of cAMP shifts the voltage activation of sea urchin SLC9C1 by −20 mV[2] and primes SLC9C1 for activation by hyperpolarization. The structural basis for priming is that the CNBD domains become more flexible, disrupting interactions with the S4 helix and decreasing cytosolic domain–domain interactions between protomers, enabling the attached ICHs to move outwards, increasing the mobility of ICH7 and altering the positioning of the VSD. In particular, the S1–S3 helices start to rotate anti-clockwise around S4. Right, on the basis of the high structural similarity between the VSD in sea urchin SLC9C1 and VSD$_{IV}$ in human Na$_V$1.7, and the fact that the mutation of the gating-charge residue R803Q reduces the elemental gating charge from 3.1 to 2.0 $e_0$ (ref. 2), we propose that S4 will probably undergo a 10 Å vertical displacement to the down-state in response to hyperpolarization. Such a large rearrangement would displace the ICH network attached to S4, therefore removing the constraints from ICH7 and enabling ion-exchange. The diagram was created using BioRender.com.

protomer (Supplementary Fig. 1 and Extended Data Fig. 8a). Indeed, the CNBD interactions, together with domain-swapped ICH1, means that the cytoplasmic domains are forming multiple interactions to one another, burying a total homodimerization interface of around 6,100 Å[2] (Figs. 3g and 4a and Supplementary Fig. 6b). Sandwiched between the NHE transporter and the extensive cytoplasmic ICH network is ICH7 (Fig. 4b), which is the only cytosolic region making multiple contacts with the NHE transporter. Specifically, Lys939 forms a salt bridge to Glu300 as well as hydrogen bonding to backbone carbonyl oxygen atoms in the end of the linker helix TM7 (Fig. 4a). ConSurf analysis[49] corroborates that a positive charge in ICH7 is strictly conserved (Supplementary Fig. 1 and Extended Data Fig. 10a). The linker helix TM7 connects the core ion-translocation domain to the dimerization scaffold domain and, in an elevator mechanism, rotates to enable core domain movement[23]. Thus, we propose that, at 0 mV, SLC9C1 is inactive because ICH7 is physically restricting core domain movement. Consistent with this interpretation, three-dimensional variability analysis (3DVA) reveals that ICH7 is flexible, with larger domain movements between the positively charged CNBD interfaces (Supplementary Video 2). States in which ICH7 is interacting most extensively with the transporter domains are associated with stronger map density for Lys939 and TM7, whereas more detached states show poorer map density and VSD mobility.

The binding of cAMP to the CNBDs shifts the activation of SLC9C1 to more positive voltages[2]. To understand the molecular basis of cAMP regulation, we determined the cryo-EM structure of SLC9C1 in the presence of cAMP, which was built into the EM maps reconstructed to 3.6 Å (Extended Data Table 1 and Extended Data Fig. 9). Consistent with SLC9C1 being inactive at 0 mV, we observed no structural differences in the NHE transporter module (Fig. 4b). However, the addition of cAMP substantially alters the stability of SLC9C1. The CNBDs could not be

modelled, with either absent or poor map features for the CNBDs at lower contour levels (Extended Data Fig. 10b), and this decreased the size of the modelled cytoplasmic interface (Fig. 4b and Supplementary Fig. 6b). The judicial placement of ICH7 was clearer in the 3D reconstruction collected with cAMP, as it is the most distal region that could be modelled, apart from a proline-rich stretch that connects ICH7 with the CNBDs (Fig. 4c, Extended Data Fig. 10c and Supplementary Fig. 1). It is well known that cAMP binding induces a clam-like shell closing of the αC helix in CNBDs[45] (Extended Data Fig. 8b). This likely rearrangement seems to have displaced polar residue interactions between the cytoplasmic end of S4 and the αC and αD helices in the CNBD (Fig. 4c and Extended Data Fig. 10b). Consistent with this interpretation, S4 and the connecting ICH3, ICH4 and ICH5 helices have moved outwards from their previously modelled position (Fig. 4c and Extended Data Fig. 10d).

The movement of the ICH3–ICH6 helices in the presence of cAMP is propagated to the VSD domains and increases their mobility (Fig. 4c). As such, cryo-EM map density for only one of the VSD domains was observed and at a lower local resolution of about 3.5 to 6.5 Å resolution (Extended Data Fig. 9b). Nevertheless, the map quality was sufficient for assigning the overall position of the S1–S4 helices (Extended Data Fig. 9c). Comparing the VSD domain in the presence and absence of cAMP showed that there is an anti-clockwise displacement of S1–S3 helices on the intracellular side of around 7.5 to 10 Å (Fig. 4c). Notably, the conformational changes observed are similar to the rotation of S1–S3 that takes place during the vertical displacement and unwinding of the S4 helix between the up- and down-states in Na$_V$1.7 channels[47]. Although Lys939 in ICH7 is still interacting with the linker helix TM7 (Extended Data Fig. 10c), 3DVA reveals similar, yet more extensive, domain movements to those seen in GDN without cAMP addition (Supplementary Videos 2 and 3). ICH7 moves together with the rest of the cytoplasmic network, including S4 and the VSD. The detachment of ICH7 from the

transporter module is clearly correlated with the repositioning of S4 and the VSD (Supplementary Video 3). Thus, it appears that cAMP has primed SLC9C1 for activation by detaching CNBD interactions with the cytoplasmic end of S4 (Extended Data Fig. 10d).

## Summary

The sperm-specific NHE SLC9C1 has evolved so that its ion-exchange activity is extrinsically regulated by voltage. Although NHEs have dedicated roles in fine-tuning cytoplasmic and organellular pH[14], SLC9C1 is thought to have a specialized role in sperm signalling by controlling the pH-dependent activation of sAC[7,2]. Here, the cryo-EM structures of SLC9C1 demonstrate that, when inactive (0 mV), the transporter module is accessible to $Na^+$ binding, which is consistent with $Na^+$-induced currents measured by SSM-based electrophysiology. However, the ion-exchange activity of the transporter module is inhibited because the intracellular ICH7 restricts the movement of the linker helix TM7, which is required for NHE transporter activity. Such an autoinhibitory model is consistent with the most recent cryo-EM structure of the $K^+/H^+$ exchanger KefC from *E. coli*[38], which is regulated by cytoplasmic RCK domains that also inhibit activity by interacting with the linker helix TM7. Moreover, the helix-turn-helix motif in the C-linker of HCN1 channels, regulating channel opening under hyperpolarization potentials, is topologically equivalent to the ICH6 and ICH7 helices proceeding the CNBD in the SLC9C1 transporter[45], although the outcome is different; that is, cAMP binding to CNBD relieves inhibition in HCN channels by promoting different domain–domain interactions[50], whereas in SLC9C1 it relives inhibition by disrupting them.

Given the close structural similarity between the VSDs in SLC9C1 and $Na_V1.7$, we expect that S4 will undergo a similar vertical displacement of around 10 Å to the down-state after hyperpolarization[42,47]. Consistently, the mutation of Arg803 (R2) to glutamine lowered the gating charge from 3.1 to 2.0 $e_0$ and, therefore, this residue is likely to cross the entire TM electric field[2]. As the ICH domains are connected only to S4, this degree of downward movement would be sufficient to break the interactions between ICH7 and the transporter module. In the absence of cAMP, a larger electrical force (voltage) is required to move the S4 helix into the down-state[2] (Fig. 5). However, the binding of cAMP to the CNBDs, appears to make it easier for the S4 helix to move downwards by (1) destabilizing S4–CNBD interactions; (2) weakening homodimerization contacts; and (3) partially detaching ICH7 and rearranging S1–S3 to facilitate S4 movement. Such a regulatory mechanism seems to share features of the VSD activation of $Na_V$ channels[42,47] and the allosteric coupling between cAMP binding and the removal of CNBD inhibition in hyperpolarization-activated HCN channels[45], yet there are obvious differences. In voltage-gated channels, the position of the VSD is directly coupled to the open- or closed-state probability of the channel through the S4–S5 linker, whereas, in the SLC9C1 transporter, the VSD is required only to remove an inhibition so that the transporter can then cycle through its conformational states independently. The major differences in electromechanical coupling reflect the functional distinction between channels and transporters—channels conduct a multitude of ions in a single open state, whereas transporters need to cycle through multiple states to translocate a single substrate ion[24].

As sperm function outside the organism, we postulate that evolution has converted a transporter into a pH transducer, rather than a proton channel, as a proton channel could equilibrate cytoplasmic pH only to that encountered in its external environment. The further coupling of the SLC9C1 transporter to a VSD module used by depolarization-activated channels would also ensure that, after activation of the CatSper $Ca^{2+}$ channel, the SLC9C1 transporter would be rapidly inactivated again. Thus, the SLC9C1 protein provides a clear example of how transporters can be adapted to function in signalling pathways. From a mechanistic viewpoint, the SLC9C1 structure highlights the plasticity of the VSD and S4, in particular, to establish alternative pathways for electromechanical coupling, which could be potentially exploited by protein-engineering strategies. Taken together, our study into the only known VSD-regulated transporter provides important molecular insights into a transporter that is essential for sperm motility and fertilization, and reveals mechanistic themes that are relevant to ion transporters and channels in general.

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

## Methods

### Expression and purification of SLC9C1

The gene encoding the *S. purpuratus* SLC9C1 protein (NP_001091927.1), referred to as SLC9C1, was synthesized (Thermo Fisher Scientific) and subcloned into the pcDNA3.1(+) vector followed by a TEV (Tobacco Etch Virus nuclear-inclusion-a endopeptidase) recognition site, eGFP and a TwinStrep tag at the C terminus. An Asp238Ala variant of SLC9C1 was further constructed using the Quickchange kit (Life Technologies). The recombinant vectors were verified by DNA sequencing and purified using the PureLink Giga Plasmid Purification Kit (Invitrogen). The serum-free suspension-adapted Freestyle HEK293F cells (Invitrogen) were used for expression and cultured in FreeStyle 293 expression medium (Life Technologies), shaken at 140 rpm in a humidified incubator at 37 °C and under a 8% $CO_2$ atmosphere. Cell growth and viability was assessed using a Countess II FL automated cell counter (Invitrogen). The cells were transiently transfected using 20 kDa linear polyethylenimine (PEI) (PolySciences). Transfections were performed using a final DNA concentration of 1 μg ml⁻¹. In brief, cloned *Slc9c1-gfp* containing pcDNA3.1(+) vector was diluted in FreeStyle medium and vortexed for 10 s. PEI-20KDa was diluted using the same medium at a 1:2 (w/w) DNA:PEI ratio and vortexed for 10 s before being added to the DNA-containing medium. The DNA:PEI mixture was incubated for 10 min at room temperature and added to the cells at a cell density of 1–1.5 × 10⁶ cells per ml. Then, 24 h after transfection, sodium butyrate was added to a final concentration of 5 mM. Typically, 48 h later 10 l of cell culture was collected by centrifugation (3,000*g* at 4 °C for 10 min). The cells were subsequently resuspended in 100 ml cell resuspension buffer (20 mM Tris-HCl pH 7.5, 150 mM NaCl) supplemented with 1 tablet of cOmplete, EDTA-free protease inhibitor cocktail (Roche) and subsequently lysed using a homogenizer. The membranes were isolated from the resulting pellet after ultracentrifugation at 195,000*g* and 4 °C for 1 h.

For structural studies using cryo-EM, the isolated membranes were solubilized in solubilization buffer containing 2% (w/v) glyco-diosgenin (GDN, Anatrace), 50 mM Tris-HCl pH 7.5, 150 mM NaCl, 0.003% brain fraction 7 (Sigma-Aldrich) and 10% glycerol for 2 h at 4 °C with mild agitation. The solution was subsequently cleared by ultracentrifugation 195,000*g* and 4 °C for 1 h. The supernatant was applied to GFP nanobody-coupled CNBr-activated Sepharose 4B resin (Cytiva) and incubated for 1 h at 4 °C, and subsequently washed with 10 column volumes (CV) of wash buffer consisting of 0.02% GDN (w/v), 20 mM Tris-HCl pH 7.5, 150 mM NaCl and 0.003% brain fraction 7 (w/v). The resin-bound GFP-TwinStrep-His₈ residue was cleaved by TEV-His₈ protease present in excess amounts during mild agitation overnight at 4 °C. The cleaved protein was collected and concentrated using 100 kDa MW cut-off spin concentrators (Amicon, Merck-Millipore). The concentrated SLC9C1 protein was subsequently injected onto a pre-equilibrated Superose 6 increase 10/300 column (GE Healthcare) in GDN SEC buffer consisting of 50 mM Tris-HCl pH 7.5, 150 mM NaCl and 0.02% (w/v) GDN, and the peak fraction at 13.6 ml was collected using the Äkta system. Owing to the low levels of produced protein, typically around 0.3 mg of purified protein could be isolated from 10 l of HEK293 cell culture.

For reconstitution of SLC9C1 into nanodiscs, SLC9C1 was purified as described above and concentrated to approximately 1 mg ml⁻¹. Concentrated SLC9C1 was subsequently mixed with yeast polar lipids (Sigma-Aldrich) and MSP1E2 nanodiscs at a molar ratio of 1:4:200 for 30 min at room temperature. To this solution, 50 mg of preactivated SM-2 biobeads (Bio-Rad) was added and allowed to incubate overnight at 4 °C. The next day, 50 mg of fresh biobeads were added and further incubated for 1 h. The biobeads were discarded and the solution was centrifuged at 14,000 rpm for 5 min. The reconstituted nanodiscs were further processed by size-exclusion chromatography on a Superose 6 increase column in a buffer containing 50 mM Tris pH 7.5 and 150 mM NaCl. Peak fractions corresponding at 13.6 ml were collected.

### SSM-based electrophysiology

For SSM-based electrophysiology measurements, purified protein in GDN SEC buffer was reconstituted into liposomes. In brief, yeast polar lipids (Avanti) were solubilized in a 2:1 (v/v) chloroform:methanol solution and thin-layered using a rotary evaporator (Hei-Vap Core, Heidolph Instruments). The lipids were thoroughly resuspended in 10 mM MES-Tris pH 7.5, 5 mM $MgCl_2$ buffer at a final concentration of 10 mg ml⁻¹. Unilamellar vesicles were prepared by extruding the resuspended lipids using an extruder (Avestin) with 400 nm polycarbonate filters (Whatman). The 500 μg of extruded lipids were destabilized by adding sodium cholate to a final concentration of 0.65% (w/v) (Sigma-Aldrich). Then, 20 μg of either the SEC-purified SLC9C1 or D238A variant was added to the destabilized liposomes at a desired LPR of 125:1 (w/w) (or, as stated in Supplementary Fig. 3, the protein ratio was tested between LPR 5 to 500) and incubated for 5 min at room temperature. Purified rat GLUT5 and the horse NHE9 ion-binding site mutant D244A/N243A were prepared as previously described[18,51] and added to destabilized liposomes at an LPR of 5:1 (w/w) and incubated for 5 min at room temperature. The samples were individually applied to the PD SpinTrap G-25 desalting column (Cytiva) to remove excess detergent and the reconstituted proteoliposomes were collected in a final volume of 500 μl. The sample was diluted to final lipid concentration of 1 mg ml⁻¹ in 10 mM MES-Tris at the desired pH, ranging from 6.5, 7.5 to 8.5 in a buffer otherwise consisting of 5 mM $MgCl_2$, 5 mM NaCl and 5 mM KCl buffer, and then flash-frozen at −80 °C. Proteoliposomes were diluted 1:1 (v/v) with non-activating buffer made up of 10 mM MES-Tris at the desired pH, 200 mM choline chloride, 5 mM $MgCl_2$ and sonicated using a bath sonicator before sensor preparation. All electrophysiology measurements were recorded using the SURFE²R N1 Nanion instrument. Sensor preparation for SSM-based electrophysiology on the SURFE²R N1 system (Nanion Technologies), was performed according to the manufacturer's instructions[30]. For symmetrical pH measurements and different LPR comparisons, 3 mm sensors were used, otherwise 1 mm sensors were used for all other experiments. In brief, sensors were incubated with 50 μl of a 0.5 mM octadecanethiol solution (Sigma-Aldrich) and kept at room temperature for 30 min. After the incubation, the sensors were washed with 100% isopropanol (Sigma-Aldrich) and deionized water. A 1 μl droplet of lipid solution (1,2-diphytanoyl-*sn*-glycero-3-ph osphatidylcholine (Avanti) in *n*-decane (Sigma-Aldrich) at a final concentration of 7.5 mg ml⁻¹) was added to the gold surface, followed by 50 μl of non-activating buffer. Then, 5 μl of the diluted proteoliposomes was added to the sensors and the sample was then centrifuged at 3,000*g* for 30 min to ensure the complete adhesion of the proteliposomes to the surface. SLC9C1 was activated by changing from non-activating buffer to an activating buffer containing the NaCl. To measure the binding kinetics, 200 mM choline chloride in non-activating buffer was replaced by *x* mM NaCl and (200−*x* mM) choline chloride in the activating buffer. The peak with the largest absolute value was recorded as the peak current. Three sensors were prepared for each sample and at least two replicate measurements were made for each sensor. Current traces were corrected for small offset differences (<50 pA). The current showed pre-steady-state ion translocation rather than steady-state. The decay time constant τ was determined using a monoexponential fit (one-phase decay) in GraphPad Prism using the slope calculated between the highest to the lowest peak plateau at around 1.3 s for each sensor.

### Lipid–protein interactions assessed by FSEC

SLC9C1 was repurified in the detergent *n*-dodecyl-β-ᴅ-maltopyranoside (DDM)/cholesteryl hemisuccinate Tris salt (CHS) without GFP cleavage. In brief, the SLC9C1–GFP-containing membranes were solubilized in solubilization buffer containing 1% (w/v) DDM (Glycon technologies), 0.2% CHS (Sigma-Aldrich), 50 mM Tris-HCl pH 7.5, 150 mM NaCl and 0.003% brain fraction 7 (w/v) for 1 h at 4 °C with mild agitation. The solution was subsequently cleared by ultracentrifugation 195,000*g*,

4 °C for 1 h. The supernatant was applied to Strep-Tactin Sepharose resin (IBA) and incubated for 1 h at 4 °C and subsequently washed with 10 CV of wash buffer consisting of 0.05% DDM (w/v), 0.01% (w/v) CHS, 20 mM Tris-HCl pH 7.5 and 150 mM NaCl. The resin-bound SLC9C1–GFP was eluted in 2 CV of elution buffer containing 0.03% DDM (w/v), 0.006% (w/v) CHS, 20 mM Tris-HCl pH 7.5, 150 mM NaCl and 1× BXT (IBA). The elution was applied to a pre-equilibrated PD-10 desalting column (Cytiva) using reaction buffer consisting of 20 mM Tris pH 6.5, 150 mM NaCl and 0.03% DDM. The samples were concentrated to a final concentration of 0.5–0.75 mg ml$^{-1}$.

To characterize the thermostability and lipid thermal stabilization of SLC9C1–GFP, ~0.05 mg of samples were incubated for 10 min at 4 °C without lipids as a control and with the individual lipids at a final concentration of 1 mg ml$^{-1}$ in reaction buffer. Stock solutions of the respective lipids (1,2-dioleoyl-$sn$-glycero-3-phosphate, sodium salt (DOPA, Avanti, 840875P), 1,2-dioleoyl-$sn$-glycero-3-phosphoethanolamine (DOPE, Avanti, 850725P), 1,2-dioleoyl-$sn$-glycero-3-phosphocholine (DOPC, Avanti, 850375P), 1,2-dioleoyl-$sn$-glycero-3-phospho-(1′-rac-glycerol), sodium salt (DOPG, Avanti, 840475P) and 1,2-dioleoyl-$sn$-glycero-3-phospho-L-serine, sodium salt (DOPS, Avanti, 840035P) were prepared by solubilization in 10% (w/v) DDM to a final concentration of 30 mg ml$^{-1}$. Subsequently, β-D-octyl-glucoside (Anatrac) was added to a final concentration of 0.1% (w/v) and the sample aliquots of 120 μl were heated for 10 min either at 4 °C or 45 °C after the addition of a final concentration of 0.33% (w/v) DDM or individual lipids in a PCR thermocycler (Veriti, Applied Biosystems). The heat-denatured material was subsequently pelleted at 18,000$g$ for 30 min at 4 °C. The supernatant was injected into the Bio-Rad ENrich 650 column, pre-equilibrated with 20 mM Tris, 150 mM NaCl and 0.03% (w/v) DDM, analysed using fluorescence-detection size-exclusion chromatography (FSEC; Shimadzu HPLC LC-20AD/RF-20A)[52,53]. The FSEC traces of purified SLC9C1–GFP were recorded without lipid addition at 4 °C and with and without lipid addition at 45 °C. Data used for bar chart comparisons were the fluorescence intensity of the FSEC dimerization peak in response to different lipids or additional DDM from three technical repeats.

### Cryo-EM sample preparation and data acquisition
**GDN sample.** For structural studies, 3.5 μl of 3 mg ml$^{-1}$ purified SLC9C1 in GDN was applied to a freshly glow-discharged Quantifoil R1.2/1.3 Cu300 grid with a blot force of 0, a wait time of 15 s wait and a blot time of 3 s using the Vitrobot Mark IV (Thermo Fisher Scientific) at 4 °C and 100% humidity, and plunge-frozen into liquid ethane. Cryo-EM datasets were collected on the Titan Krios G2 electron microscope operated at 300 kV equipped with a GIF (Gatan) and a K3 BioQuantum direct electron detector (Gatan) in counting mode. The video stacks were collected at ×130,000, corresponding to a pixel size of 0.6645 Å at a dose rate of 13–14 e$^-$ per physical pixel per second. The total exposure time for each video was 1.8 s, leading to a total accumulated dose of 58.45 e$^-$ Å$^{-2}$ and was fractionated into 40 frames. All videos were recorded with a defocus range of −0.6 to −2.0 μm

**Nanodisc sample.** For structural studies, 3.5 μl of 1 mg ml$^{-1}$ SLC9C1 in nanodiscs was blotted on quantifoil 2/1 grids with a blot force of 0, a wait time of 5 s and a blot time of 3 s using the Vitrobot Mark IV maintained at 4 °C and 100% humidity. Cryo-EM datasets were collected on the Titan Krios G2 electron microscope operated at 300 kV equipped with a GIF (Gatan) and a K3 BioQuantum direct electron detector (Gatan) in counting mode. The video stacks were collected at ×130,000 corresponding to a pixel size of 0.6645 Å at a dose rate of 14.37 e$^-$ per physical pixel per second. The total exposure time for each video was 1.8 s, leading to a total accumulated dose of 58.61 e$^-$ Å$^{-2}$ and was fractionated into 40 frames. All videos were recorded with a defocus range of −0.6 to −2.0 μm.

**GDN sample with cAMP.** Prior to blotting, a final concentration of 0.1 mM cAMP and 0.1 mM MgCl$_2$ was added to 3 mg ml$^{-1}$ of SLC9C1 purified in GDN. A sample volume of 3.5 μl was blotted onto the Quantifoil R 2/1 grid with a blot force of 0, a wait time of 5 s and a blot time of 3 s using the Vitrobot Mark IV maintained at 4 °C and 100% humidity. Cryo-EM datasets were collected on the Titan Krios G2 electron microscope operated at 300 kV equipped with a GIF (Gatan) and a K3 BioQuantum direct electron detector (Gatan) in counting mode. The video stacks were collected at ×130,000 magnification corresponding to a pixel size of 0.6645 Å at a dose rate of 13.30 e$^-$ per physical pixel per second. The total exposure time for each video was 1.7 s, leading to a total accumulated dose of 51.21 e$^-$ Å$^{-2}$ and was fractionated into 40 frames. All videos were recorded with a defocus range of −0.6 to −2.0 μm. All statistics of cryo-EM data acquisition are summarized in Extended Data Table 1.

### Image processing
**GDN sample.** The dataset for the GDN sample was processed by CryoSparc[54]. Dose-fractionated video frames were aligned using patch motion corrections, and the contrast transfer function (CTF) was estimated using 'patch CTF estimation. Exposures with an estimated CTF fit resolution of lower than 5.0 Å were discarded. Automated particle picking was performed using a blob picker from 3,000 random images to generate 2D templates for template-based particle picking. A total of 2,039,463 particles picked with a box size of 512 pixels were extracted and Fourier-resampled to 128 pixels. After several rounds of 2D classification, 312,610 particles were used for ab initio model building followed by several rounds of non-uniform refinement. 3D variability cluster analysis was performed to generate five clusters, which were used as the input for further rounds of hetero refinement applying $C_2$ symmetry. A final map was produced by performing local refinement at 3.23 Å. To improve the map density of the VSD, the final particle stack was expanded using symmetry expansion. Local refinement was performed using a monomer mask which improved the overall density and generated gold standard FSC resolution estimation at 3.2 Å. To further improve the density of VSD, local refinement was performed using a mask encompassing only the VSD and cytosolic domains.

**Nanodisc sample.** The dataset for the nanodisc sample was processed by cryoSparc[54]. Does-fractionated video frames were aligned using patch motion corrections, and the CTF was estimated using patch CTF estimation. Exposures with an estimated CTF fit resolution worse than 4.0 were discarded. Automated particle picking was performed using a blob picker from 500 random images to generate 2D templates for template-based particle picking. A total of 1,737,146 particles were picked and extracted with a box size of 550 pixels and Fourier-resampled to 400 pixels. After several rounds of 2D classification, 276,031 remaining particles were processed for multimodel ab initio model building followed by several rounds of hetero refinements using $C_2$ symmetry. 3D variability cluster analysis was performed to generate three clusters, which were used as the input for further rounds of hetero refinement applying $C_2$ symmetry. A final map was produced by performing non-uniform refinement at 3.08 Å containing 97,279 particles. To improve the density for the VSD domain, an intermediate map with $C_2$ symmetry produced after 3DVA in cryoSPARC[54] and hetero refinement was used to produce five additional ab initio classes. From this, a class with the best features for the VSD domain was processed for a round of hetero refinement along with a $C_2$ map to segregate particles with clearer density for the VSD domain. After further rounds of hetero refinement in $C_1$ symmetry, the best volume was processed for sequential masked local refinement, first by masking out the density for the nanodisc, followed by masking out the density for one monomer corresponding to a poor VSD to finally yield a map with enhanced features for a single monomer with its corresponding VSD at an overall resolution of 3.30 Å containing 100,697 particles.

**GDN and cAMP sample.** The dataset for the GDN sample in the presence of cAMP was processed in CryoSPARC[54]. Dose-fractionated video frames were aligned using patch motion corrections and CTF was estimated using patch CTF estimation. Exposures with an estimated CTF fit resolution worse than 5.0 were discarded. Particles were picked using an automated blob picker with a picking diameter of 180 Å. Picked particles were extracted with a box size of 448 pixels and Fourier-cropped to 128 pixels and subsequently used for several rounds of 2D classification. Particles were then rescaled to a box size of 240 pixels and used for ab initio maps. The best ab initio model along with 'junk' models were processed for several rounds of heterogeneous refinement. Particles corresponding to the best resulting map were rescaled to a box size of 320 pixels and processed for a round of non-uniform refinement in $C_1$ resulting in a map with an overall resolution of 3.68 Å containing 94,091 particles. 3DVA of the final reconstruction was performed in cryoSPARC[54].

## Model building and refinement of SLC9C1 in GDN, nanodiscs and GDN with cAMP

The SLC9C1 homology model was taken from AlphaFold[55] and each domain was extensively refitted into the $C_2$ GDN map using the fit in map utility of Chimera[56] and rebuilt extensively in Coot[57]. The structure was refined using real-space refinement in Phenix[58]. The side chains for the VSD were built in Coot[57] using the focused refinement maps obtained after symmetry expansion and masked local refinement, shown in Supplementary Video 1. The built SLC9C1 VSD monomer was refitted into the $C_2$ maps of the SLC9C1 homodimer and processed for rigid-body refinement using Phenix[58]. The final refinement statistics are shown in Extended Data Table 1. For model building of the SLC9C1 structure in nanodiscs, the SLC9C1 structure in GDN was first fitted into the $C_2$ nanodisc cryo-EM map using the fit in map utility of Chimera[56] and manually adjusted in Coot[57]. As the nanodisc cryo-EM map had weak density for the VSDs, the side chains were trimmed at the Cβ position in the final model. The SLC9C1 nanodisc structure was refined with real-space refinement in Phenix[58]; the final refinement statistics are shown in Extended Data Table 1. For model building of the SLC9C1 structure in GDN with cAMP, the apo SLC9C1 structure in GDN was fitted into the cryo-EM map using the fit in map utility of Chimera[56]. The structure was manually inspected and trimmed to remove regions that were not supported by cryo-EM map density. Iterative model building into the map was performed using Coot[57], and the structures were refined using real-space refinement by Phenix[58]; final refinement statistics are shown in Extended Data Table 1. Density for VSDs was poor in the GDN maps with cAMP and, as such, only one VSD domain was built with residues trimmed at the Cβ position. To illustrate the structure and cryo-EM maps, PyMol[59] and Chimera[56] were used.

## Reporting summary

Further information on research design is available in the Nature Portfolio Reporting Summary linked to this article.

## Data availability

The coordinates and the maps for SLC9C1 have been deposited at the PDB and Electron Microscopy Data Bank (EMD) under accession codes 8OTQ and EMD-17182 (SLC9C1 in GDN), 8OTX and EMD-17186 (SLC9C1 in nanodiscs) and 8OTW and EMD-17185 (SLC9C1 in GDN and cAMP). Source data are provided with this paper.

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

**Acknowledgements** Figs. 1a and 5 were created using BioRender. We thank G. von Heijne, M. Claesson and J. Cowgill for feedback on the manuscript; A. Bazzone for discussions on SSM-based electrophysiology data; and the members of the Cryo-EM Swedish National Facility at SciLifeLab Stockholm for cryo-EM data collection, funded by the Knut and Alice Wallenberg Foundation, the Family Erling Persson and Kempe Foundations and SciLifeLab. This work was funded by a European Research Council (ERC) Consolidator Grant EXCHANGE (ERC-CoG-820187) to D.D.

**Author contributions** D.D. designed the project. Cloning, expression screening and sample preparation for cryo-EM were performed by H.Y. and V.M. Cryo-EM data collection and map reconstruction were performed by H.Y., V.M. and A.G. Model building was performed by V.M., D.D. and A.G. Experiments for functional analysis were performed by H.Y. All of the authors discussed the results and commented on the manuscript.

**Funding** Open access funding provided by Stockholm University.

**Competing interests** The authors declare no competing interests.

**Additional information**
**Correspondence and requests for materials** should be addressed to David Drew.

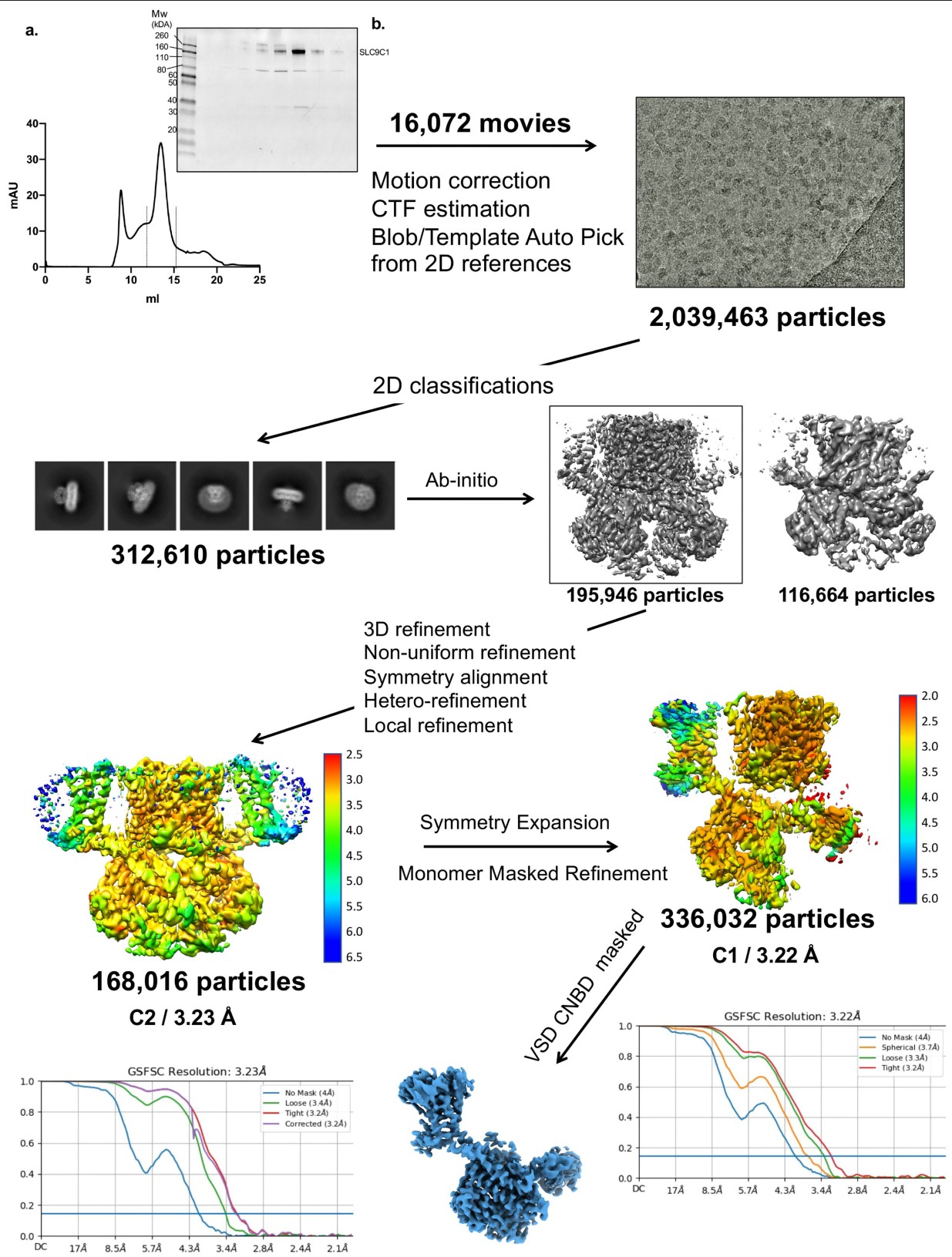

**Extended Data Fig. 1** | See next page for caption.

**Extended Data Fig. 1 | The data-processing workflow of *sea urchin* SLC9C1 in GDN. a**. Size-exclusion chromatography trace of *Sea urchin* SLC9C1 after nanobody GFP-affinity based purification in GDN and on-column TEV cleavage and removal; eleven independent purifications were performed with similar results. The arrow indicates the collected SLC9C1 protein peak collected for blotting as analysed by SDS-page and Coomassie staining (inset). **b**. The data-processing workflow of *sea urchin* SLC9C1 in GDN. a. The dataset contained 16,072 videos that were corrected by patch motion correction and patch CTF estimation in cryoSPARC[54]. After reference-based auto-picking, 2,039,463 particles were picked. Several rounds of 2D classification were performed, yielding 312,610 particles, which were subjected to 3D classification. One of the 3D classes was selected, and it contained 195,956 particles. After repeated heterogenous and local refinement led to a final resolution of 3.23 Å in C2 symmetry at gold-standard FSC (0.143), with a local resolution range of 2.5–6.5 Å. To improve the density of the VSDs, symmetry expansion in C1 and masked refinement was further applied (light-blue maps) with the model-to-map fitting shown in Extended Data Fig. 3 and Supplementary Video 1. A final resolution of 3.22 Å was achieved at gold-standard FSC (0.143), with a local resolution range of 2.0–6.0 Å.

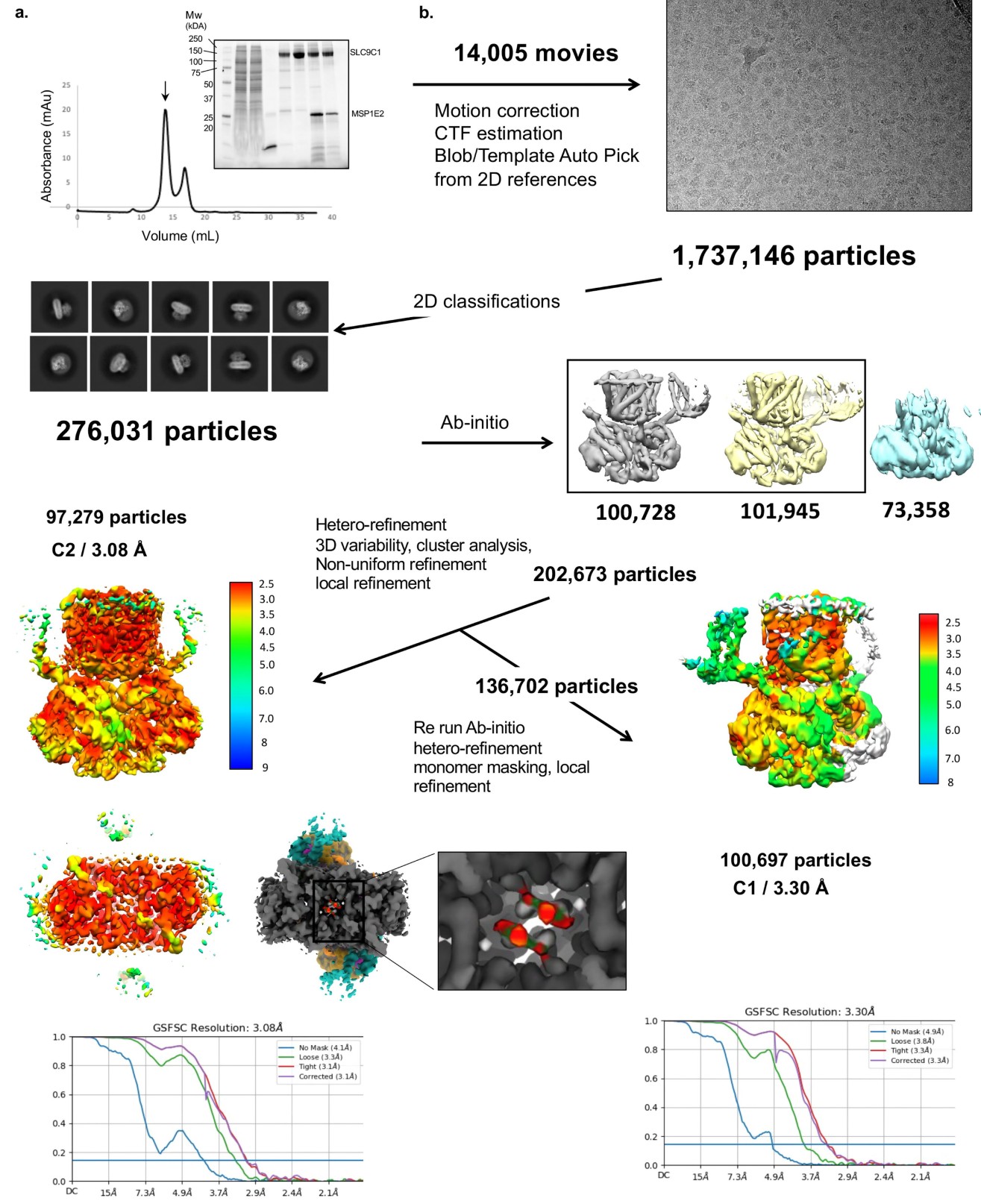

**Extended Data Fig. 2** | See next page for caption.

**Extended Data Fig. 2 | The data-processing workflow of *sea urchin* SLC9C1 in nanodiscs. a**. Size-exclusion chromatography trace of *Sea urchin* SLC9C1 after nanobody GFP-affinity based purification in GDN and reconstitution into MSP1E2 nanodiscs; triplicate independent purifications were performed with similar results. The arrow indicates the collected SLC9C1 protein peak collected for blotting as analysed by SDS-page and Coomassie staining is shown (inset). **b**. The data-processing workflow of *sea urchin* SLC9C1 in nanodiscs a. The dataset contained 14,005 movies that were corrected by patch motion correction and patch CTF estimation in cryoSPARC[54]. After reference-based auto-picking, 1,737,146 particles were picked. Several rounds of 2D classification were performed, yielding 276,031 particles, which were subjected to 3D classification. Two of the 3D classes was selected, and after repeated heterogenous and local refinement led to a final resolution of 3.08 Å in C2 symmetry at gold-standard FSC (0.143), with a local resolution range of 2.5–9.0 Å. Maps as coloured in Fig. 1f. highlight that the lipid density was resolved to a similar local resolution of the protein density at 2.5 to 3.0 Å resolution. To improve the map quality for the VSDs, 136,702 particles from the C2 processing were subjected to a further round of *ab-initio* 3D reconstruction, followed by hetero refinement in C1 symmetry. The nanodisc and monomer densities were sequentially masked resulting in a final C1 map at an overall lower resolution of 3.30 Å, but with improved map quality for one of the VSDs.

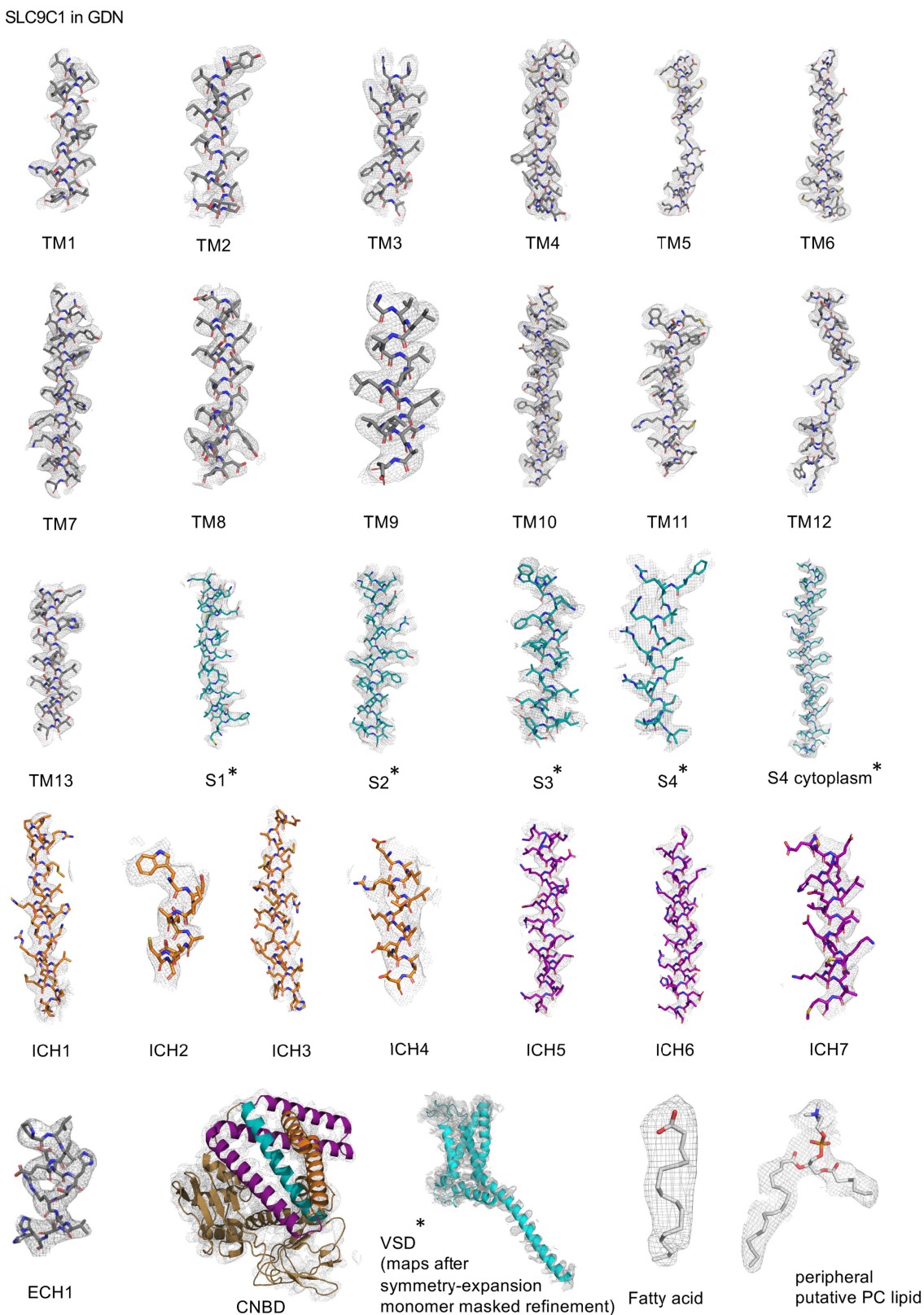

SLC9C1 in GDN

TM1  TM2  TM3  TM4  TM5  TM6

TM7  TM8  TM9  TM10  TM11  TM12

TM13  S1*  S2*  S3*  S4*  S4 cytoplasm*

ICH1  ICH2  ICH3  ICH4  ICH5  ICH6  ICH7

ECH1  CNBD  VSD
(maps after
symmetry-expansion
monomer masked refinement)*  Fatty acid  peripheral
putative PC lipid

**Extended Data Fig. 3 | Cryo-EM density and model of SLC9C1 in GDN.**
C2 symmetry cryo-EM maps from SLC9C1 in GDN (grey-mesh) and model are shown for all the transmembrane segments of the NHE transporter module (grey), ICH1-4 (orange), ICH5-7 (purple). The CNBD domain (sand), the putative lipid PC (grey sticks) are shown as full domains with cryo-EM map in grey mesh. The cryo-EM density maps in GDN after symmetry-expansion and monomer-masked refinement (annotated with *) used to refine the subsequent model for S1-S4 of the VSD (cyan) are further shown.

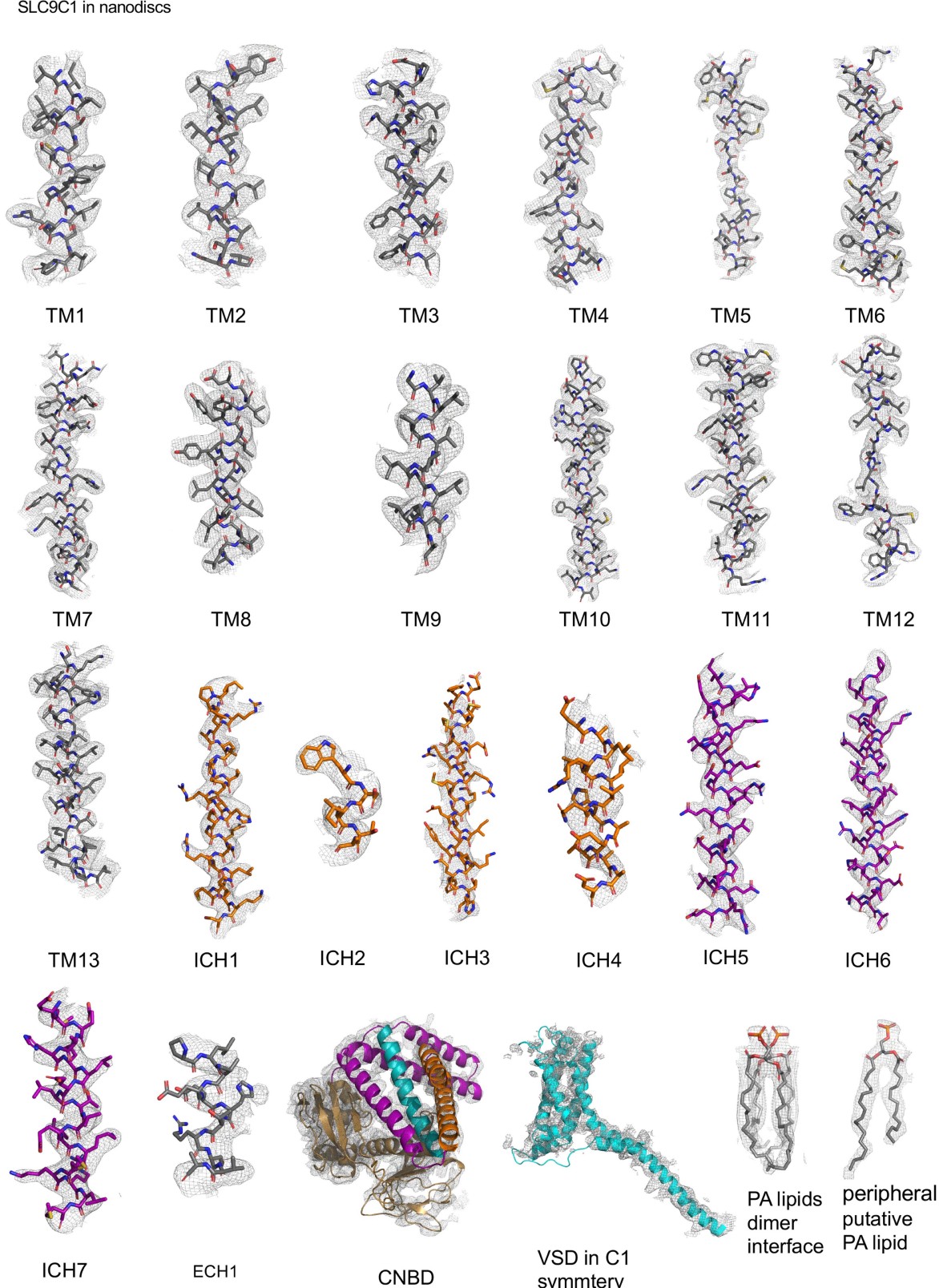

SLC9C1 in nanodiscs

TM1  TM2  TM3  TM4  TM5  TM6

TM7  TM8  TM9  TM10  TM11  TM12

TM13  ICH1  ICH2  ICH3  ICH4  ICH5  ICH6

ICH7  ECH1  CNBD  VSD in C1 symmtery  PA lipids dimer interface  peripheral putative PA lipid

**Extended Data Fig. 4 | Cryo-EM density and model of SLC9C1 in nanodiscs.** C2 symmetry cryo-EM maps from SLC9C1 in nanodiscs (grey-mesh) and model are shown for all the transmembrane segments of the NHE transporter module (grey), ICH1-4 (orange), ICH5-7 (purple). The CNBD domain (sand), the PA lipids at the dimerization interface (grey-sticks) and a peripheral putative PA lipid (grey-sticks) are further shown. C1 symmetry cryo-EM maps used for fitting the VSD domain (cyan) is further shown, but no side-chain density was modelled in S1-S4 due to weak cryo-EM map density.

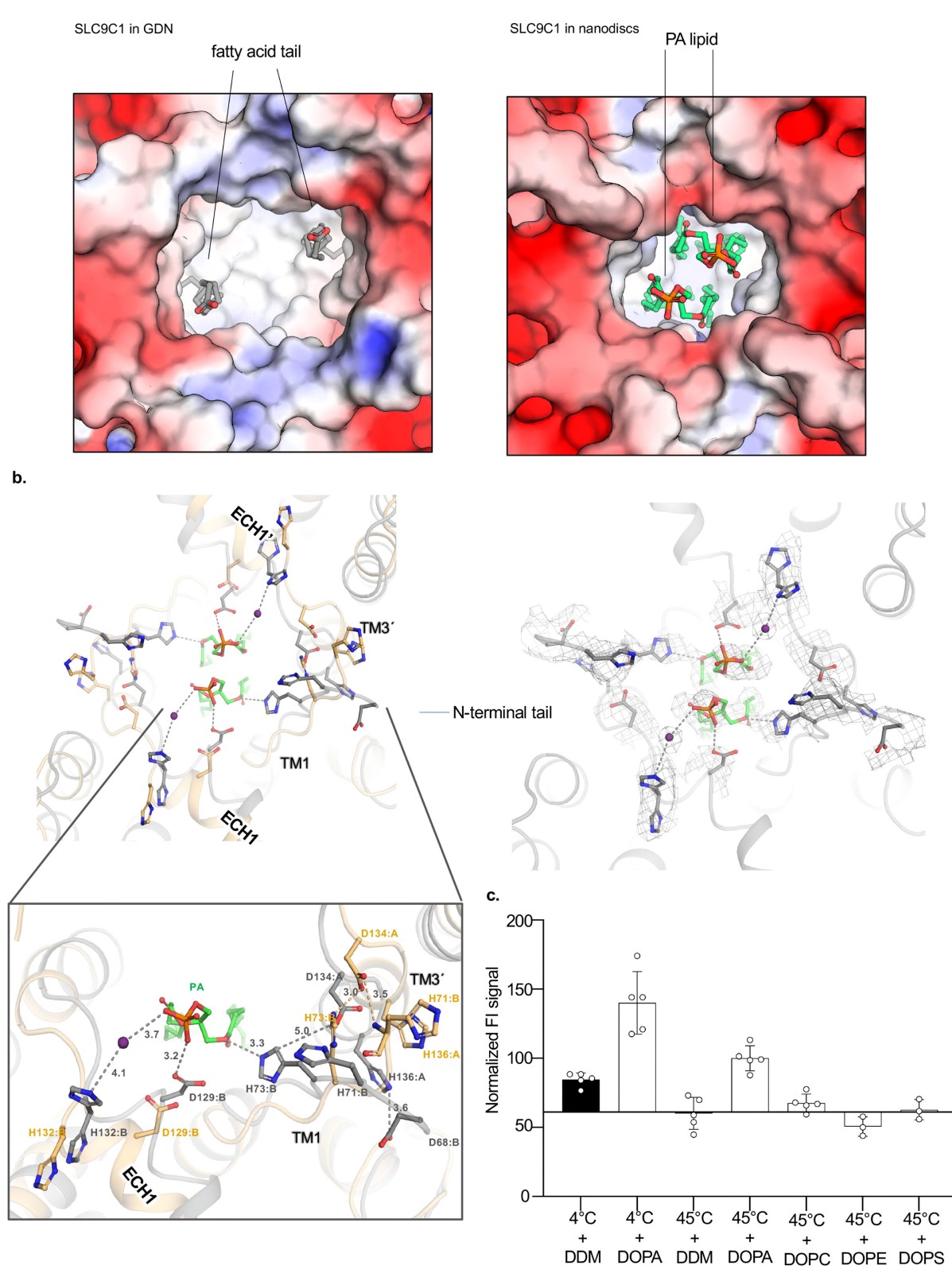

a.

SLC9C1 in GDN

fatty acid tail

SLC9C1 in nanodiscs

PA lipid

b.

ECH1′

TM3′

ECH1

TM1

N-terminal tail

D134:A

D134:A

3.0

3.5

H71:B

PA

3.7

H73:B

3.2

5.0

H136:A

3.3

H73:B

H136:A

H71:B

H132:B

H132:B

4.1

D129:B

D129:B

3.6

D68:B

ECH1

TM1

c.

Normalized FI signal

200

150

100

50

4°C
+
DDM

4°C
+
DOPA

45°C
+
DDM

45°C
+
DOPA

45°C
+
DOPC

45°C
+
DOPE

45°C
+
DOPS

**Extended Data Fig. 5** | See next page for caption.

**Extended Data Fig. 5 | Lipid-induced structural changes at the dimeric interface. a**. Electrostatic surface of the SLC9C1 homodimer in detergent (left) and in nanodiscs (right) as viewed from the extracellular side, and bound fatty acid tails (grey-sticks) and PA lipids (green-sticks) are labelled **b.** Left: cartoon representation of the SLC9C1 structure of the dimerization interface in detergent (light brown) and in nanodiscs made with crude yeast polar lipids (grey). The N-terminal tail of TM1 extends across to the opposite-facing protomer and forms interactions with the extracellular end of TM3'. In the detergent structure (light-brown) H73B forms a salt-bridge with D134 as well as a backbone hydrogen bond to H71B, as shown in the zoomed in view below. In the nanodisc structure (grey), cryo-EM map density supporting two well-resolved PA lipids were able to be modelled (green sticks). To coordinate the PA lipid, the phosphate head-group hydrogen bonds to D129 in ECH1, most probably its uncharged state, as well as to H132 *via* a water-molecule, likely in the −1 charged state. The carbonyl oxygen from the fatty acid ester further hydrogen bonds to H73 in the N-terminal tail form the opposite-facing protomer. The intricate coordination of the negatively-charged PA lipids in the nanodisc SLC9C1 structure has stabilized a more compacted homodimer than in detergent. Other $Na^+/H^+$ exchangers have proposed to bind specific lipids at the oligomerization interface and are thought to regulate oligomerization and functional activity[17,18,33,52,60]. Notably, the phosphomonoester headgroup of PA is the only lipid headgroup to alter its charged state around physiological pH values, and is thus also been referred to a pH-sensing lipid[61]. Right: as in the left panel showing the nanodisc structure and corresponding cryo EM map density (grey mesh) **c**. Quantification of the normalized homodimer FSEC peak height of SLC9C1-GFP after incubation of DDM/CHS purified fusion with either DDM or various DDM-solubilized lipids and heating (see Methods). Representative FSEC traces are shown in Supplementary Fig. 6c. Error bars are the mean value ± $n$ = 5 independent experiments for SLC9C1 at 4 °C, at 45 °C, at 4 °C plus DOPA, at 45 °C plus DOPC, at 45 °C plus DOPC and mean value ± $n$ = 3 independent experiments for 45 °C plus DOPE, 45 °C plus DOPS.

**a.**

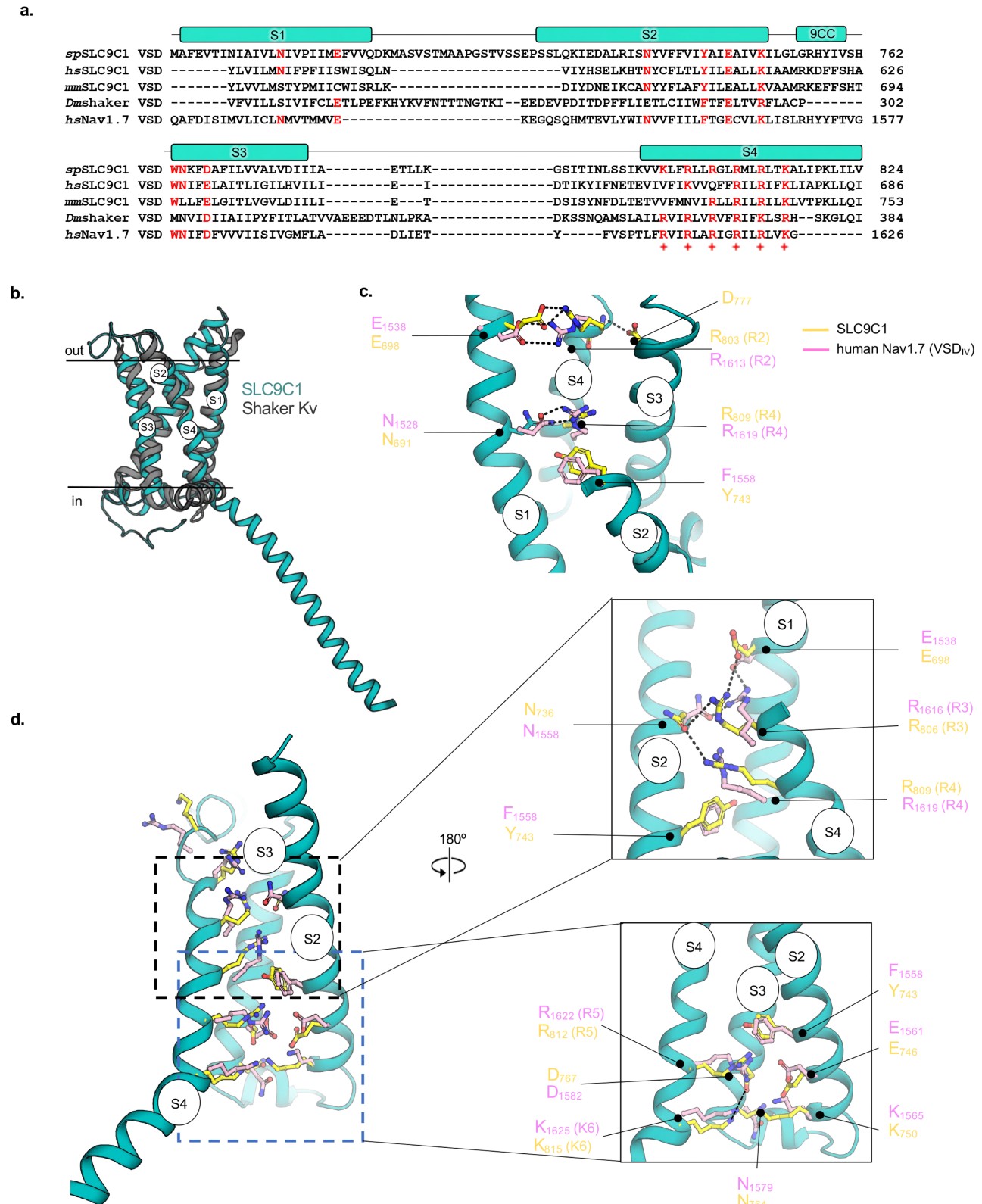

| | | S1 | | | S2 | | 9CC | |
|---|---|---|---|---|---|---|---|---|

*sp*SLC9C1 VSD  MAFEVTINIAIVL**N**IVPIIM**E**FVVQDKMASVSTMAAPGSTVSSEPSSLQKIEDALRIS**N**YVFF**VIY**AIE**A**IVKILGLGRHYIVSH  762
*hs*SLC9C1 VSD  -------YLVILM**N**IFPFIISWISQLN--------------------VIYHSELKHT**N**YCFLTL**YI**LE**A**LLKIAAMRKDFFSHA  626
*mm*SLC9C1 VSD  -------YLVVLMSTYPMIICWISRLK--------------------DIYDNEIKCA**N**YYFLAF**YI**LE**A**LLKVAAMRKEFFSHT  694
*Dm*shaker VSD  -------VFVILLSIVIFCL**E**TLPEFKHYKVFNTTTNGTKI--EEDEVPDITDPFFLIETLCIIW**F**TFE**LTV**R**FLACP------- 302
*hs*Nav1.7 VSD  QAFDISIMVLICL**N**MVTMMV**E**--------------------KEGQSQHMTEVLYWI**N**VVFIIL**F**TGE**CVL**KLISLRHYYFTVG 1577

| | | S3 | | | | | | S4 | | |
|---|---|---|---|---|---|---|---|---|---|---|

*sp*SLC9C1 VSD  **WN**KF**D**AFILVVALVDIIIA--------ETLLK--------------GSITINLSSIKVV**K**LF**R**LL**R**GL**R**ML**R**LT**K**ALIPKLILV  824
*hs*SLC9C1 VSD  **WN**IF**E**LAITLIGILHVILI-------E---I--------------DTIKYIFNETEVIVF**I**KVVQFF**R**IL**R**IF**K**LIAPKLLQI  686
*mm*SLC9C1 VSD  **W**LLF**E**LGITLVGVLDIILI-------E---T--------------DSISYNFDLTETVVFMNV**I**RLL**R**IL**R**IL**K**LVTPKLLQI  753
*Dm*shaker VSD  MNV**I**D**IIAIIPYFITLATVVAEEEDTLNLPKA--------------DKSSNQAMSLAIL**R**VI**R**LV**R**VF**R**IF**K**LS**R**H--SKGLQI  384
*hs*Nav1.7 VSD  **WN**IF**D**FVVVIISIVGMFLA-------DLIET--------------Y-----FVSPTLF**R**VI**R**LA**R**IG**R**IL**R**LV**K**G-------  1626
                                                                     +  +  +  +  +  +

**b.**

**c.**

**d.**

**Extended Data Fig. 6** | See next page for caption.

**Extended Data Fig. 6 | Sequence and structural comparison with VSD in SLC9C1 versus VSD in voltage-gated ion channels. A**. Clustal omega[62] protein sequence alignment of the VSD domain of *sea urchin* SLC9C1, *human* SLC9C1 (uniport: Q4G0N8), *mouse* SLC9C1 (Q6UJY2), *drosophila* K⁺-shaker channel (P08510), and $VSD_{IV}$ of *human* sodium channel $Na_V$1.7 (Q15858). Conserved residues are highlighted in red and gating-charge residues in S4 are further highlighted with a + symbol. **b**. Superimposition of the SLC9C1 VSD (cyan, labelled S1 to S4) with VSD from the Shaker Kv K⁺-channel (PDB:7SIP, grey) with an r.m.s.d 2.4 Å for all $C_\alpha$ pairs. **c**. Cartoon representation of the VSD in the *sea urchin* SLC9C1 structure in detergent (cyan) highlighting the conserved salt-bridge and hydrogen bond interactions between R2 and R4 residues in S4 and E698 and N691 in S1 (yellow-sticks), respectively. The equivalent conserved residues in $VSD_{IV}$ of *human* $Na_V$1.7 (PDB: 7XMG) are further shown (pink sticks). The aromatic residue Y743 (F1558 in $Na_V$1.7) in the charge transfer centre is also highlighted. **d**. Cartoon representation of the gating-charge residues in SLC9C1 (yellow sticks) and $Na_V$1.7 (pink sticks) as well as the charge transfer centre. Inset above: The gating-charge residue R806 in SLC9C1 forms a hydrogen-bond to N736 in S2 and salt bridged to E698 in S1 as in *human* $Na_V$1.7 (pink sticks). The gating-charge residue R809 in SLC9C1 also forms a hydrogen bond to N736 in S2. Inset below: Apart from the gating aromatic residue, all the residues in the charge transfer centre are conserved and similarly positioned between the VSD in *sea urchin* SLC9C1 (yellow sticks) and $VSD_{IV}$ of human $Na_V$1.7 (pink sticks).

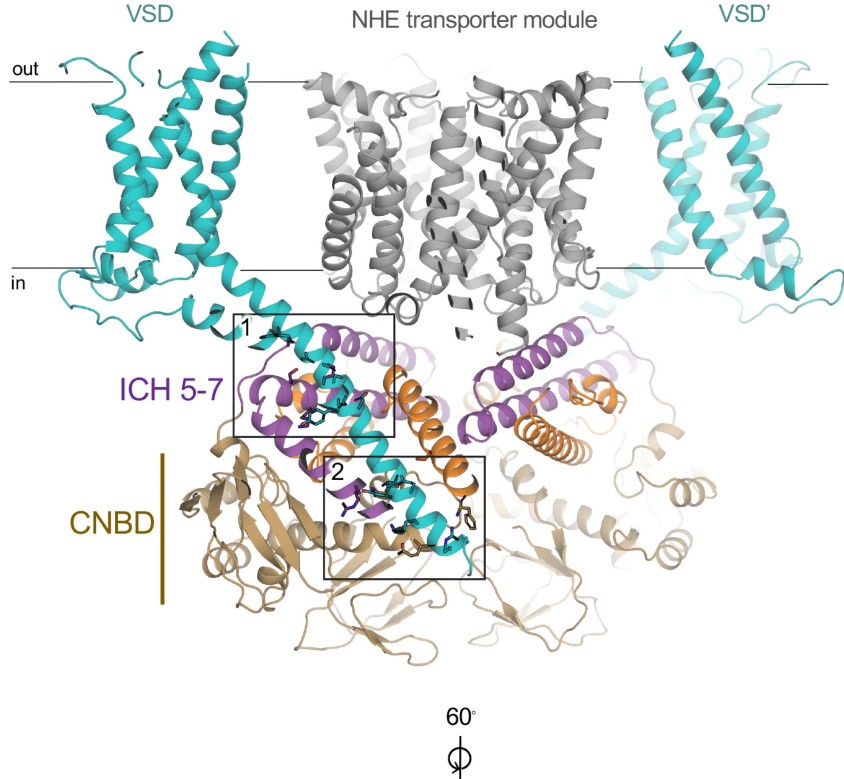

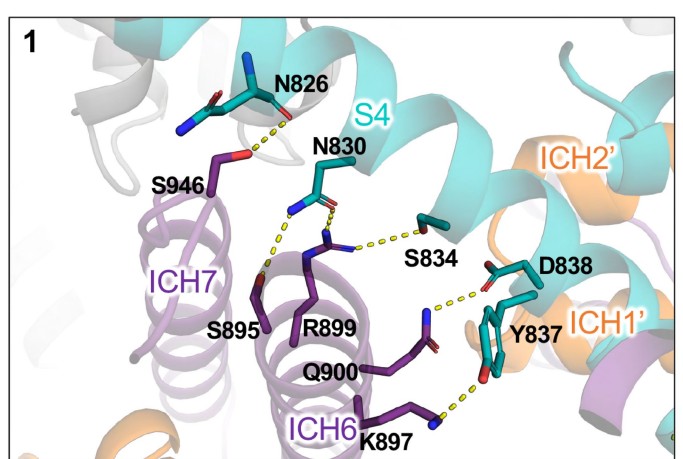

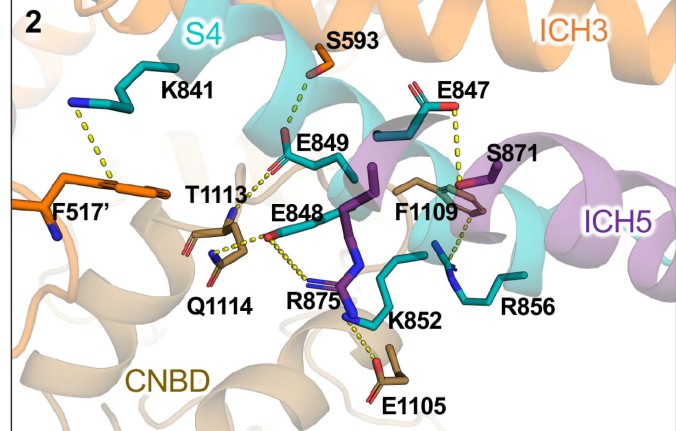

**Extended Data Fig. 7 | Polar interactions between extra membranous S4 and ICHs and the CNBDs.** *top*: Cartoon representation of the SLC9C1 homodimer, with the NHE transporter (grey), VSD domains (teal) and CNBD (browns) and connecting intracellular helices ICH1-ICH4 (orange) and ICH5-ICH7 (purple). *bottom-left:* left, zoomed in view highlighting the polar interactions (1) between the start of the extra membranous region in S4 (teal-sticks) with ICH6 and the start of ICH7 (purple sticks). Yellow dashed-lines show salt-bridge and hydrogen-bonding interactions up to 4 Å. *bottom-right*, zoomed in view highlighting the polar interactions (2) between the cytoplasmic end of the extra membranous region in S4 (teal sticks) with ICH3 (orange-sticks), ICH5 (purple-sticks) and CNBD helices (brown-sticks). Yellow dashed-lines show salt-bridge and hydrogen-bonding interactions up to 4 Å.

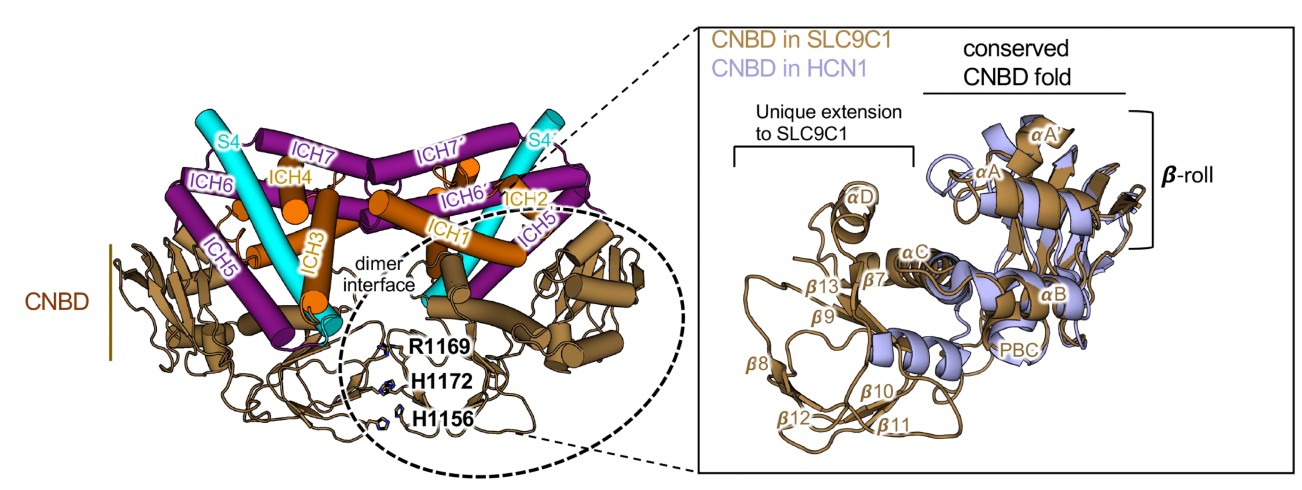

**Extended Data Fig. 8** | See next page for caption.

**Extended Data Fig. 8 | Structural comparison of the cyclic nucleotide binding domain (CNBD) in SLC9C1 and in CNBD-gated ion-channels.**
**a**. Focused view of the interactions between CNBD domains (brown) that harbour positively charged residues H1156, R1169, and H1172 (sticks). Inset right: superposition of the CNBD in *sea urchin* SCL9C1 (brown) and the CNBD in the *human* HCN1 channel (blue, PDB ID: 5U6P). The main structural difference is that the CNBD in SLC9C1 has an additional helix (αD) and a β-roll domain (β7-β13), which extends the CNBD-structural fold to form unique homodimerization contacts. **b**. left, structural superimposition of the *apo* CNBD in the HCN2 channel (PDB: 2MPF, light-blue) and *apo* CNBD in SLC9C1 (brown). right, structural superimposition of the cAMP-bound CNBD in the HCN2 channel (PDB: 3U10, dark-blue) and apo CNBD in SLC9C1 (brown). Inset below: cAMP in HCN2 (dark-pink sticks) is coordinated by residues (green sticks) that are mostly conserved in SLC9C1 (yellow sticks). The coordination of cAMP in HCN2 stabilizes the closed state of the αC helix as labelled. **c**. left, structural superimposition of the *apo* CNBD in the MloK1 channel (PDB: 2KXL, dark-pink) and *apo* CNBD in SLC9C1 (brown). right, structural superimposition of the cAMP-bound CNBD in the MloK1 channel (PDB: 2KOG, light-green) and *apo* CNBD in SLC9C1 (brown). Inset below: cAMP in MloK1 (grey sticks) is coordinated by residues (pink sticks) that are mostly conserved in SLC9C1 (yellow sticks). The coordination of cAMP in MloK1 stabilizes the closed state of the αC helix as labelled.

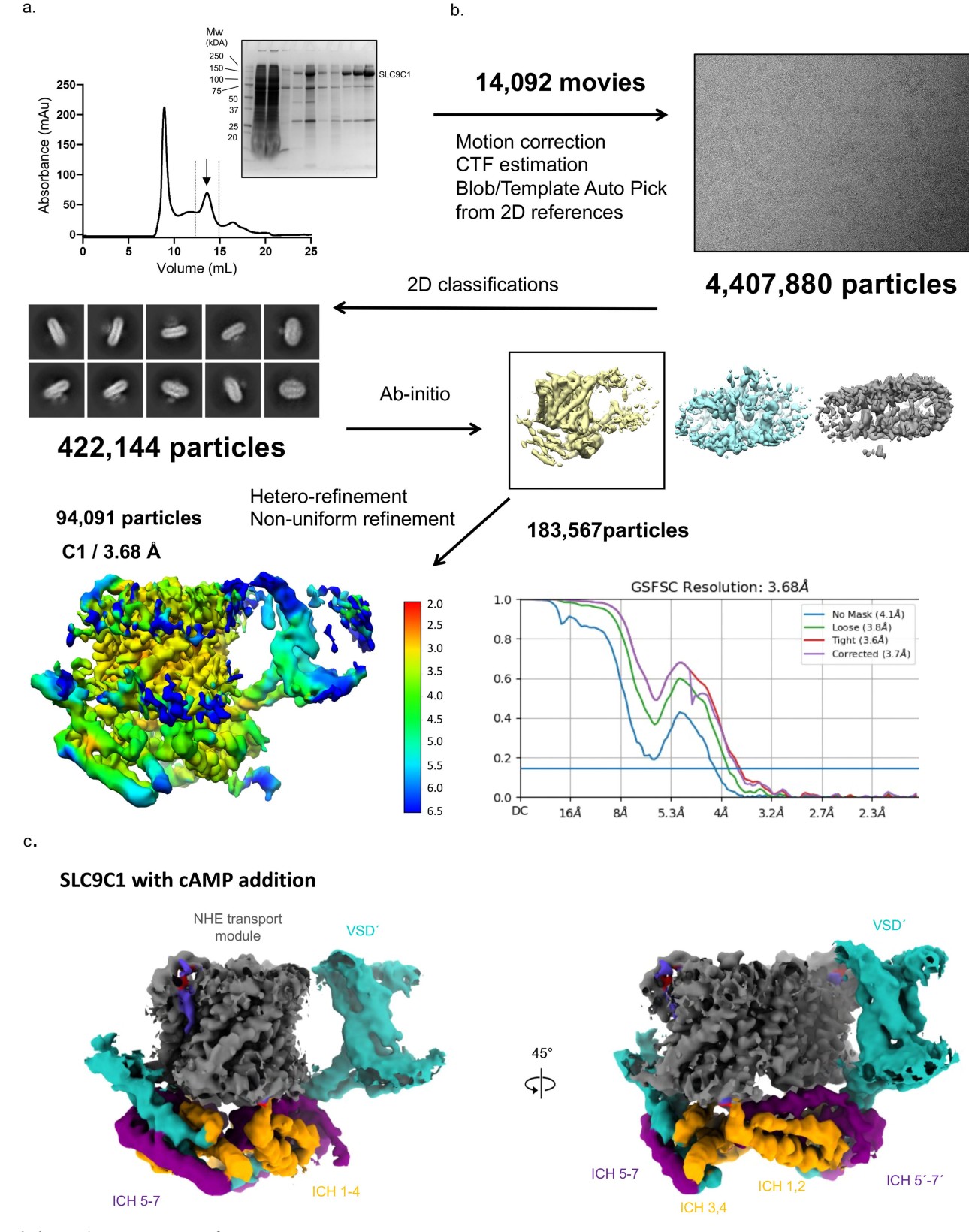

a.

14,092 movies

Motion correction
CTF estimation
Blob/Template Auto Pick
from 2D references

b.

4,407,880 particles

2D classifications

422,144 particles

Ab-initio

Hetero-refinement
Non-uniform refinement

183,567particles

94,091 particles

C1 / 3.68 Å

c.

**SLC9C1 with cAMP addition**

NHE transport module

VSD´

ICH 5-7

ICH 1-4

45°

VSD´

ICH 5-7

ICH 1,2

ICH 3,4

ICH 5´-7´

**Extended Data Fig. 9** | See next page for caption.

**Extended Data Fig. 9 | Cryo-EM density and model of SLC9C1 in GDN with the addition of cAMP. a**. Size-exclusion chromatography trace of *sea urchin* SLC9C1 after nanobody GFP-affinity based purification and on-column TEV cleavage and removal; eleven independent purifications were performed with similar results. The arrow indicates the collected SLC9C1 protein peak collected for blotting in GDN with 0.1 mM cAMP addition as analysed by SDS-page and Coomassie staining (inset) **b**. The data-processing workflow of *sea urchin* SLC9C1 in GDN with 1 mM cAMP. The dataset contained 14,092 movies that were corrected by patch motion correction and patch CTF estimation in cryoSPARC[54]. After reference-based auto-picking, 4,407,880 particles were picked. Several rounds of 2D classification were performed, yielding 424,144 particles, which were subjected to 3D classification. One of the 3D classes was selected, and after repeated heterogenous and local refinement led to a final resolution of 3.68 Å in C2 symmetry at gold-standard FSC (0.143), with a local resolution range of 2.0–6.5 Å. **c**. Maps as coloured in Fig. 1f. highlight that the VSD (cyan) in the SLC9C1 structure with cAMP addition was only apparent for one of the VSD domains, and the local map resolution was not sufficient to build side-chains. No map density was present for the CNBD domains.

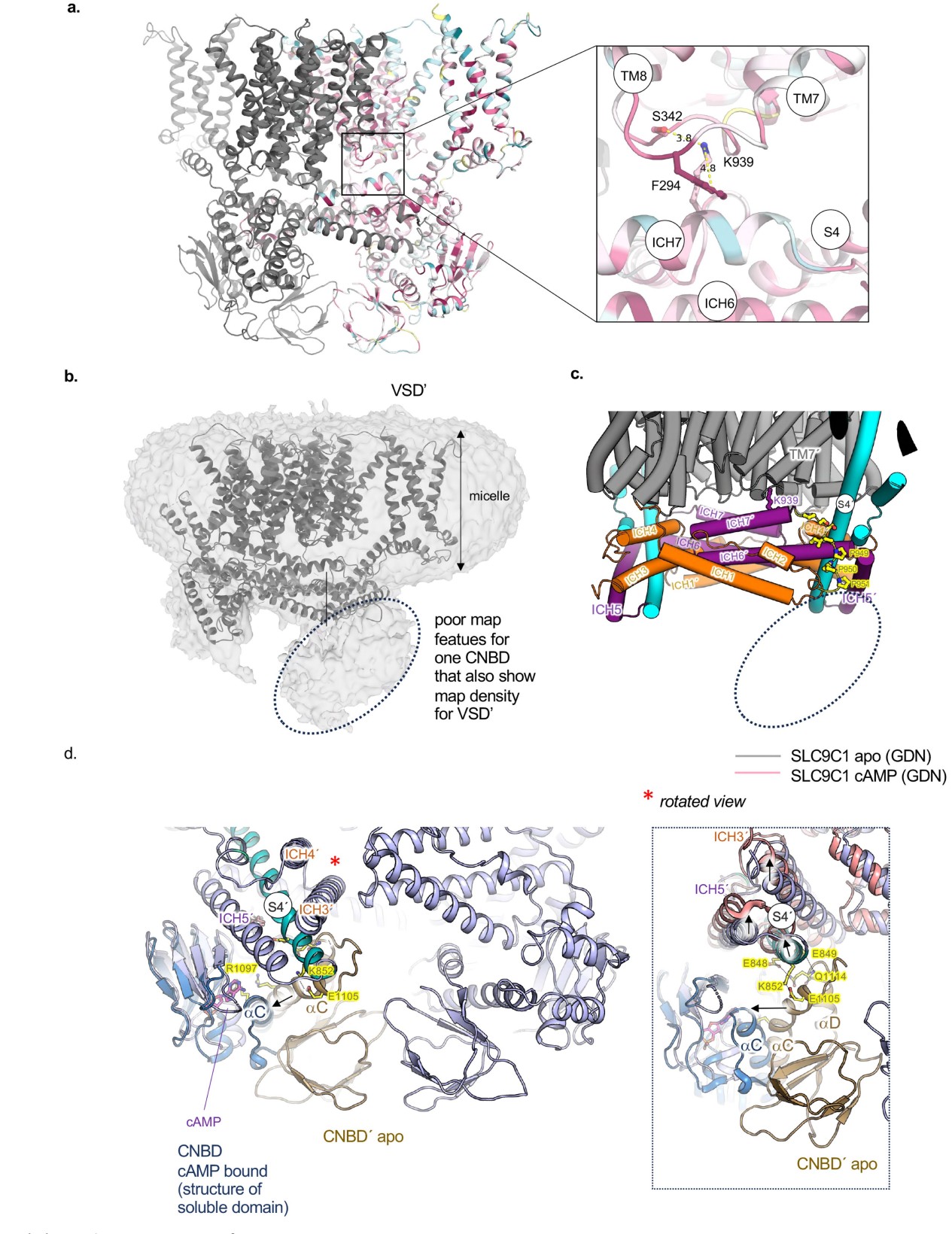

**Extended Data Fig. 10** | See next page for caption.

**Extended Data Fig. 10 | ConSurf of ICH7 and cAMP induced differences in SLC9C1. a**. ConSurf[63] analysis from the sequence alignment in Supplementary Fig. 1, which was based on SLC9C1 homologues belonging to different phylla. The conservations scores are shown on the structure of one monomer from blue (non-conserved) to dark-red (strictly-conserved). Inset: zoomed in view of the ICH7 lysine residue (K939), which is invariant with either arginine or less-frequently histidine. The K939 residue mostly forms backbone interactions, but also a weak hydrogen bond to S342 and π-cation interactions to highly conserved residue F294 in TM7. **b**. At a low map threshold, there is cryo EM map density (grey surface) for one of the CNBDs (dotted-ellipse), but the resolution is too poor to accurately position this domain. **c**. Cartoon of the modelled SLC9C1 structure with cAMP addition showing that ICH7′ maintains interactions with TM7′ at 0 mV. A rigid proline-rich linker (yellow-sticks) that connects ICH7′ to the now "absent" CNBD domain (dotted-ellipse) is further highlighted. **d**. Cartoon representation of the cytosolic region of SLC9C1 (tinted-grey) with a focused view of the extramembranous region of S4 (teal) and interactions to residues (yellow sticks, labelled) in αC and αD helices (brown) in the CNBD domain (tinted grey) of SLC9C1. The superimposition of the cAMP-bound (dark-pink, sticks) in the CNBD from the HCN2 channel (PDB: 2MPF, dark-blue), highlights that the inward movement of αC in SLC9C1 would break interactions to S4. Inset right: A zoomed in view and rotation of the view from the cytoplasmic side to highlight the salt-bridge and hydrogen-bond interactions (dashed-line) between the end of S4 (teal) and residues in αC and αD helices of the CNBD (brown). Consistent with cAMP stabilizes closure of the αC helix, in the cAMP-bound structure, S4 and connecting intracellular helices (pink) have moved away from the CNBD.

**Extended Data Table 1 | Cryo-EM data collection, refinement and validation statistics**

|  | SLC9C1 – GDN (EMDB- 17182) (PDB 8OTQ) | SLC9C1 in nanodisc EMDB- 17186 PDB 8OTX | SLC9C1- cAMP EMDB- 17185 PDB 8OTW |
|---|---|---|---|
| **Data collection and processing** | | | |
| Magnification | | 130,000 | |
| Voltage (kV) | | 300 | |
| Electron exposure (e–/Å$^2$) | 58.45 | 58.61 | 51.21 |
| Defocus range (μm) | | -0.6 to -2.0 | |
| Pixel size (Å) | | 0.6645 | |
| Symmetry imposed | C2 | C2 | C1 |
| Initial particle images (no.) | 2,039,463 | 1,737,146 | 4,333,119 |
| Final particle images (no.) | 168,016 | 97,279 | 94,091 |
| Map resolution (Å) | 3.22 | 3.08 | 3.68 |
| FSC threshhold | | 0.143 | |
| Map resolution range (Å) | 3.16 – 3.95 | 3.00 - 3.76 | 3.56 – 4.23 |
| **Refinement** | | | |
| Model resolution (Å) | 3.5 | 3.28 | 3.90 |
| FSC threshold | 0.5 | 0.5 | 0.5 |
| **Model composition** | | | |
| Non-hydrogen atoms | 16716 | 16116 | 11571 |
| Protein residues | 2102 | 2108 | 1522 |
| Ligands | 2CPL, 2PLM | 3PH | CPL |
| *B* factors (Å$^2$) | | | |
| Protein | 151.87 | 154.30 | 141.11 |
| Ligand | 127.70 | 131.42 | 120.03 |
| R.m.s. deviations | | | |
| Bond lengths (Å) | 0.005 | 0.003 | 0.003 |
| Bond angles (°) | 0.802 | 0.442 | 0.697 |
| **Validation** | | | |
| MolProbity score | 2.44 | 2.44 | 1.88 |
| Clashscore | 15.03 | 35.07 | 13.11 |
| Poor rotamers (%) | 2.9 | 0.13 | 0.09 |
| Ramachandran Plot | | | |
| Favored (%) | 94.10 | 93.83 | 96.29 |
| Allowed (%) | 5.8 | 6.17 | 3.71 |
| Disallowed (%) | 0.1 | 0.00 | 0.00 |

CPL : 1-palmitoyl-2-linoleoyl-sn-glycero-sn-3-phosphocholine
PLM : palmitic acid
3PH : 1,2-diacylglycerol3-s-n-phosphate

Table showing the cryo-EM data collection, refinement and validation statistics for the *sea urchin* SLC9C1 structure in GDN (left column), *sea urchin* SLC9C1 structure in nanodiscs (middle column) and SLC9C1 structure in GDN with cAMP (right column).

# Reporting Summary

## Statistics

For all statistical analyses, confirm that the following items are present in the figure legend, table legend, main text, or Methods section.

| n/a | Confirmed | |
|---|---|---|
| ☐ | ☒ | The exact sample size (*n*) for each experimental group/condition, given as a discrete number and unit of measurement |
| ☐ | ☒ | A statement on whether measurements were taken from distinct samples or whether the same sample was measured repeatedly |
| ☒ | ☐ | The statistical test(s) used AND whether they are one- or two-sided<br>*Only common tests should be described solely by name; describe more complex techniques in the Methods section.* |
| ☒ | ☐ | A description of all covariates tested |
| ☒ | ☐ | A description of any assumptions or corrections, such as tests of normality and adjustment for multiple comparisons |
| ☐ | ☒ | A full description of the statistical parameters including central tendency (e.g. means) or other basic estimates (e.g. regression coefficient) AND variation (e.g. standard deviation) or associated estimates of uncertainty (e.g. confidence intervals) |
| ☒ | ☐ | For null hypothesis testing, the test statistic (e.g. *F*, *t*, *r*) with confidence intervals, effect sizes, degrees of freedom and *P* value noted<br>*Give P values as exact values whenever suitable.* |
| ☒ | ☐ | For Bayesian analysis, information on the choice of priors and Markov chain Monte Carlo settings |
| ☒ | ☐ | For hierarchical and complex designs, identification of the appropriate level for tests and full reporting of outcomes |
| ☒ | ☐ | Estimates of effect sizes (e.g. Cohen's *d*, Pearson's *r*), indicating how they were calculated |

*Our web collection on statistics for biologists contains articles on many of the points above.*

## Software and code

Policy information about availability of computer code

| Data collection | Thermo Scientific EPU Software v3.2.0.4776 |
|---|---|
| Data analysis | Prism 7 - for data plotting and analysis<br>Phenix v1.20.1-4487 - Structural refinement software suite<br>PyMol v2.5.2 - Molecular graphics software<br>Coot v0.9.6 EL- Structural model building<br>cryoSPARC v3.3.2<br>Chimera X 1.5 |

For manuscripts utilizing custom algorithms or software that are central to the research but not yet described in published literature, software must be made available to editors and reviewers. We strongly encourage code deposition in a community repository (e.g. GitHub). See the Nature Portfolio guidelines for submitting code & software for further information.

## Data

Policy information about availability of data

All manuscripts must include a data availability statement. This statement should provide the following information, where applicable:

- Accession codes, unique identifiers, or web links for publicly available datasets
- A description of any restrictions on data availability
- For clinical datasets or third party data, please ensure that the statement adheres to our policy

The 3D cryo-EM density (EMD) maps and PDB coordinates have been deposited with the following accession codes:
SLC9C1 in nanodics; PDB ID: 8OTX, EMD-17186
SLC9C1 in GDN with cAMP; PDB ID 8OTW, EMD-17185
SLC9C1 in GDN; PDB ID 8OTQ, EMD-17182

## Research involving human participants, their data, or biological material

Policy information about studies with human participants or human data. See also policy information about sex, gender (identity/presentation), and sexual orientation and race, ethnicity and racism.

| | |
|---|---|
| Reporting on sex and gender | N/A |
| Reporting on race, ethnicity, or other socially relevant groupings | N/A |
| Population characteristics | N/A |
| Recruitment | N/A |
| Ethics oversight | N/A |

Note that full information on the approval of the study protocol must also be provided in the manuscript.

# Field-specific reporting

Please select the one below that is the best fit for your research. If you are not sure, read the appropriate sections before making your selection.

☒ Life sciences          ☐ Behavioural & social sciences          ☐ Ecological, evolutionary & environmental sciences

For a reference copy of the document with all sections, see nature.com/documents/nr-reporting-summary-flat.pdf

# Life sciences study design

All studies must disclose on these points even when the disclosure is negative.

| | |
|---|---|
| Sample size | Biochemical assays and were typically performed at least in triplicate (n =3) to ascertain accurate values for data shown. Statistical methods were not used to determine sample size, but the number of repeats were chosen based on standard practice in the transporter biochemistry community, and were sufficient to calculate standard deviations or standard error. |
| Data exclusions | No data was excluded. |
| Replication | All biochemical assays were repeated at least 3 times and the results were reproduced each time. |
| Randomization | Randomization was performed for calculating the Fourier-shell correlation of half-maps. |
| Blinding | No blinding was carried out for biochemical and structural analysis as this is not applicable as the analysis does not include subjects. |

# Reporting for specific materials, systems and methods

We require information from authors about some types of materials, experimental systems and methods used in many studies. Here, indicate whether each material, system or method listed is relevant to your study. If you are not sure if a list item applies to your research, read the appropriate section before selecting a response.

## Materials & experimental systems

| n/a | Involved in the study |
|---|---|
| ☒ | ☐ Antibodies |
| ☐ | ☒ Eukaryotic cell lines |
| ☒ | ☐ Palaeontology and archaeology |
| ☒ | ☐ Animals and other organisms |
| ☒ | ☐ Clinical data |
| ☒ | ☐ Dual use research of concern |
| ☒ | ☐ Plants |

## Methods

| n/a | Involved in the study |
|---|---|
| ☒ | ☐ ChIP-seq |
| ☒ | ☐ Flow cytometry |
| ☒ | ☐ MRI-based neuroimaging |

# Eukaryotic cell lines

Policy information about cell lines and Sex and Gender in Research

| | |
|---|---|
| Cell line source(s) | HEK(293)-F cells purchased from ThermoFisher scientific |
| Authentication | Cell lines were authenticated by the manufacturer and no further authentification was performed |
| Mycoplasma contamination | Cell lines were not tested for Mycoplasma contamination |
| Commonly misidentified lines (See ICLAC register) | None used |

