## [Peer Review File · Nature]

Manuscript Title: Structure and electromechanical coupling of the voltage-gated Na⁺/H⁺ exchanger SLC9C1

Reviewer Comments & Author Rebuttals

Reviewer Reports on the Initial Version:

Referees' comments:

Referee #1 (Remarks to the Author):

This is a fascinating manuscript describing the first structures of the voltage-activated SLC9C1 Na/H antiporter. This unique antiporter is expressed in sperm flagella where it plays a key role in alkalinizing the cytoplasm and activating CatSper and driving chemotaxis. Previous sequence analysis and functional studies had shown that this transporter contains a functional S1-S4 voltage-sensing domain and intracellular cyclic nucleotide binding domain (CNBD) that allosterically control Na/H exchange, but the structural basis of these mechanisms remained unknown. Here, Yeo and colleagues present the first cryo-EM structures of this intriguing SLC9C1 transporter. Three structures are presented for the *S. purpuratus* SLC9C1, two for the apo protein, either in detergent and lipid nanodiscs at 0 mV at 3.2 Å and 3 Å resolution, respectively, and then a third in detergent in the presence of cAMP at lower resolution to provide a framework for how the protein might work. The protein forms dimers, with two transport domains positioned at dimer interface with the S1-S4-like domains in the periphery. The apo structures are similar and show how the S1-S4 domain forms a voltage-sensing domain with all the essential features known from extensive studies on related voltage-activated cation channels, but instead of coupling with a pore forming domain the S4 helix projects outside the membrane into the intracellular regions to insert into a tunnel formed between a series of intracellular helices and the CNBD. In these apo structures, the transport domains are in an inward facing conformation similar to previously seen in many other transporters adopting the NhaA fold with conserved Na binding sites deep within the TM regions and open to the intracellular side of the membrane. The dimer interface contains a hydrophobic cavity seen in related transporters and in the nanodisc structure phosphatidic acid lipids are bound near the interface. In previous functional studies cAMP diminished the extent of hyperpolarization needed to activate the transporter, so the authors reasoned that structures with cAMP bound at 0 mV might provide information on the nature of the conformational change involved in activation of the transporter. In the presence of cAMP, density for the CNBD is not seen, but the authors discern movements within the ICHs and S4 that lead them to propose that membrane hyperpolarization induces an inward movement of S4 to diminish interactions between the intracellular domain and freeing the transport module to exchange Na and protons. In effect, the proposal is that inward movements of S4 diminish an inhibitory interaction between the cytoplasmic domain and the transport domain, thus enable the transport domain to exchange Na and protons. This mechanism is presented as an initial framework for thinking about the overall mechanism, but it is the apo structures that provide the most valuable foundation for understanding this hybrid family of cyclic nucleotide and voltage regulated transporters. Overall, the study appears to be of very high quality, the exposition lucid and well-written and the conclusions nuanced and for the most part are appropriately cautious. The following are suggestions for improving the manuscript in revision.

1) To improve the resolution of VSD, have the authors tried or considered using a tight mask around the VSD to do 3D Classification without image alignment after particle expansion? This might result in higher resolution and show different conformations of the VSD.

2) It would be help for the authors to add figures highlighting the interactions between S4 and the ICH and CNBD. What kinds of residue interactions are present in this region and what might be

said about how they would need to rearrange if we assume S4 moves perpendicular to the membrane by about 15-20 Å?

3) In the initial submission, it is unclear to what extent the data obtained in the presence of cAMP adds to the story given the limited resolution and missing density for the CNBDs. It would help if the authors could add a supplementary figure showing a comparison between the cryo-EM maps and models in the absence and presence of cAMP so the reader can have a better sense of what can be concluded at this stage. Is there a reason the authors used a relatively low concentration of cAMP (0.1 mM) as opposed to the higher concentrations used in ref 5 for functional studies or from related work on HCN channels? Can the authors provide more information on how cAMP influences the biochemical properties of the SLC? Also, the SDS-PAGE gel shown in Supplementary Figure 11 reveals a band between 25 and 37 kDa, which makes one wonder whether the protein has undergone proteolytic cleavage and that is why density for the CNBD is missing. A similar band is seen in Supplementary Figure 3 but that is for the sample in nanodiscs and that band was assigned to MSP1E2. The samples in Supplementary Figure 11 are in GDN and that part of the gel is not shown for the apo protein in GDN in Supplementary Figure 2. Even if this part of the manuscript can be strengthened, the authors should be quite clear that any proposed mechanism for how S4 movement controls the transporter are speculative and would require considerably more extensive efforts.

4) The experiments reported in Supplementary Figure 7C where the authors examine the influence of lipids on the thermal stability of the protein seem problematic. In these experiments, FSEC is used to detect the stability of protein after adding lipid and heating. However, the total amount of protein (aggregation + dimer + monomer) in each condition is not equal. For example, the sample without heating should have the biggest area, but it's much smaller compared to that of sample in heat + DDM + DOPA, for example. I am not sure what these data say or how important they are for the study, but this needs to be better controlled and how heat might influence the fluorophore needs to be addressed.

5) The supported bilayer experiments are a reasonable addition, but as the authors acknowledge they only demonstrate that the protein is capable of binding Na ions in a pH dependent fashion. The authors state that these experiments do not demonstrate that the expressed and purified protein is capable of transport, but they could also remind the reader that the protein construct they used for structure determination has been shown to be fully functional when expressed in mammalian cells (ref 5).

6) The structure of the voltage-sensing domain in SLC9C1 is most similar to that found in Nav1.7, so it makes sense that the Nav channel is compared most extensively, but most of the foundational work on voltage sensing comes from work on Shaker Kv channels and in places it might be worth making that connection. A structure of Shaker is now available and the best estimates of charge per channel were done on that protein (PMIDs 8663993 and 8663992), for example.

7) Lines 131 and 132 could be more clearly explained and I assume the amount of protein in these experiments is fixed?

8) Line 289 ICH3-ICH3 should be ICH3-ICH4, correct? It looks like the intracellular extension of the S4 helix 'tunnels' into the intracellular domain between the ICH helices and the CNBD. The authors might consider using that description if it is indeed correct.

9) Line 329 typo "semes"

10) The authors often use words like above or below to orient readers when it would probably be easier for the general audience if the authors used terms like internal/external or intracellular/extracellular to describe relative relationships.

11) Pg 8 line 210 at the end I think "other" is missing after "each".

12) CNB domain and CNBD are used interchangeably, and it would be good to stick with one, for example by first defining CNBD and then using that nomenclature throughout.

13) some of the blue colors in Fig 1 are a bit challenging to distinguish and other choices might make it easier to comprehend.

Referee #2 (Remarks to the Author):

The manuscript by Yeo et al presents several novel structures of a sperm-specific sea urchin hyperpolarization-activated and cAMP-modulated Na/H antiporter that contains, in addition to the transporter core, both voltage-sensor and cyclic nucleotide-binding domains. The structures are solved in both detergent and nanodiscs in the presence and absence of cAMP, they are of high resolution, and they support an exciting novel mechanism for how cAMP and voltage work together to modulate the function of transporters in this family. The S4 transmembrane helix of the voltage sensor domain, which is of extended length compared to S4 helices in other channels (~60 Å beyond the membrane) is clamped between two sets of helix-turn-helix structures that are part of an extended network of helices that forms a tight dimer interface beneath the transporter domain dimer. These helices connect to the cyclic nucleotide-binding domains and additional structures that further contribute to the dimer interface. The authors propose that at 0 mV, the helical network is sterically blocking the core domain conformational changes required for transport (ICH7 blocks TM7 movement). Addition of cAMP leads to increased dynamics in the CNBDs (so much so that the density for CNBDs is no longer defined in the cAMP-bound structure), perturbs the dimer interface and the helical network (that obstructed transporter motions) enough so that when a hypothetical downward movement of the long S4 upon hyperpolarization occurs, the inhibition of the transporter by the helical network would be removed and the transporter becomes active. The authors perform a thorough comparative analysis of this novel quirky transporter with other transporters in the family, and with voltage-gated channels (for the voltage sensor) and CNBD-containing channels (for the CNBD). The structures are impressive, the manuscript is well-written and presented, the figures are beautiful. I have only a few minor points, listed below, and a concern regarding the functional assays.

Concerns:

To assess the functionality of their purified transporter, the authors perform SSM-based electrophysiology and they state that their current measurements likely measure the binding of Na to the transporter and not transport activity, which the authors claim it is supported by increase in peak current with increase in Na concentration and no change in the time constant of decay when they decrease the lipid-to-protein ratio (LPR) (Fig.1). However, neither Fig. 1d, e, or c show that Na binding is mediated by the transporter, which is concerning. Non-specific binding to the bilayer is consistent with the peak current going up with Na concentration. The lower signal from the D238A mutant in only one Na concentration is not very compelling because very little detail is given of this mutant (sample concentration and quality, LPR, function, etc) and there is only one Na concentration tested. Fig. 1c inset shows only a normalized signal and the authors should definitely show the peak currents here, since the signal should change predictably with changes in LPR. Furthermore, the authors should state the number of biological repeats performed here (number of protein preparations tested).

Lines 320-323: Just because the CNBs cannot be modelled, one should not assume that the entire interface disappears, as the authors do. At the very least, an exact reduction in the value of buried surface area should not be stated.

I did not see a good comparison between the CNBD of the SLC9 and that of other CNBD-containing ion channels such as HCN, in various activated, resting, etc states. Overlay (better than that in Supplementary Fig. 10) would be useful.

Supplementary Fig 4a labels a PC lipid in the GDN structure. However, there is no other supporting evidence suggesting that the lipid is a PC. Unless the authors have further evidence, I recommend removing the label. They can speculate it is a PC in the text, but not label it as such.

Referee #3 (Remarks to the Author):

Summary

The manuscript by Yeo and colleagues describes the first high-resolution cryo-EM structure of the sperm-specific voltage-gated Na⁺/H⁺ exchanger SLC9C1, using the sea urchin gene as source material. The structure of this transporter is of considerable interest physiologically and to the broader pharmaceutical industry as loss-of-function of SLC9C1 in mammals impairs sperm motility and fertilization and thus a prime target for the development of male contraceptives.

Main Comments:

1. This study is original and the manuscript is well written. The experimental design is sound, the data are compelling and the conclusions are generally sound, with some reservations described below (in Other Comments). While structures for other Na⁺/H⁺ exchangers from prokaryotes and eukaryotes, including mammals, have been reported previously, what makes the structure of this transporter unique is the presence of a membrane voltage-sensing domain (VSD) and a cytoplasmic cyclic nucleotide (cAMP) binding domain (CNBD), features unparalleled amongst Na⁺/H⁺ exchangers as well as other secondary active transporters. Based on the structure, the authors provide significant biological insight into the mechanics of the VSD – a domain that shares homology to the VSD of the human voltage-gated Na⁺ channel Nav1.7 - as well as partial insight into the dynamics and role of cAMP binding to the CNBD and their impact on Na⁺/H⁺ exchange. Overall, this is a very meritorious study that advances our understanding of this structurally diverse family of cation/proton exchangers and should be of broad interest to the scientific community and society in general.

Listed below are other comments that require attention, clarification and/or experimentation.

Other Comments

1. Sea Urchin should be included in the title to indicate the biological source of the structure.
2. Please clarify the statement on lines 125-126 "In contrast to SLC9A and SLC9B members that display maximal activity at pH 8.5..." The basis for this statement is unclear as the activities of all SLC9A and SLC9B members assessed to date are driven by the relative concentration gradients of Na⁺ and H⁺, and thus are more active at high [Na⁺] and acidic pH rather than alkaline pH. I could find no information in the cited review (Ref. 11) indicating that SLC9A and SLC9B members are maximally active at alkaline pH 8.5.
3. In Figure 1d, why does the decay of the peak current at the different [Na⁺] and pH 6.5 not return to baseline in contrast to data shown in Fig. 1c?
4. In the text (lines 134-135 and 207-209) and Figure 1e, the authors indicate that sea urchin SLC9C1 Asp238 (TM6) is a strictly conserved aspartate residue in Na⁺/H⁺ exchangers that is essential for ion-binding and transport. Indeed, an Asp residue is conserved at the equivalent position in all members of the SLC9A family, and the authors back this assertion by showing that a D238A mutation in sea urchin SLC9C1 negates Na⁺ dependent currents (Figure 1e). However,

what is perplexing is that D238 is not conserved in SLC9C1 from other species, such as humans and platypus which possess a Ser (Ser179) and Ala (Ala230), respectively, at the equivalent position (see sequence alignments in supplementary Fig. 1.). How can Asp be critical if Ser and Ala are tolerated in other species. Please comment.

In regard to the above, the authors should provide some western blot data showing the protein expression of D238A, and evidence that the equivalent amounts were incorporated into the proteoliposomes for functional measurements.

5. Lines 325-328: The authors refer to a highly-conserved proline-rich linker (949-PPPPP-953) that connects ICH7 with the CNBD's, yet according to the alignment in supplementary Fig. 1, this sequence appears to be present only in sea urchin SLC9C1, but not SLC9C1 from other species. Can this really be referred to as "highly-conserved" and hence is it biologically significant?

6. Line 956: states that R399 is in TM12, but in the supplementary Fig. 1 alignment, R399 is located in TM11.

7. In the legend to supplementary Fig. 1, please define the acronym labels, e.g. ECH1, ICH1-7 etc...

John Orłowski

Author Rebuttals to Initial Comments:

Structure and electromechanical coupling of the sperm-specific voltage-gated Na⁺/H⁺ exchanger SLC9C1

Corresponding authors:

David Drew

We appreciate the positive response concerning our manuscript. We have carefully examined each remark and responded to all points below.

Referees' comments:

Referee #1 (Remarks to the Author):

This is a fascinating manuscript describing the first structures of the voltage-activated SLC9C1 Na/H antiporter. This unique antiporter is expressed in sperm flagella where it plays a key role in alkalinizing the cytoplasm and activating CatSper and driving chemotaxis. Previous sequence analysis and functional studies had shown that this transporter contains a functional S1-S4 voltage-sensing domain and intracellular cyclic nucleotide binding domain (CNBD) that allosterically control Na/H exchange, but the structural basis of these mechanisms remained unknown. Here, Yeo and colleagues present the first cryo-EM structures of this intriguing SLC9C1 transporter. Three structures are presented for the *S. purpuratus* SLC9C1, two for the apo protein, either in detergent and lipid nanodiscs at 0 mV at 3.2 Å and 3 Å resolution, respectively, and then a third in detergent in the presence of cAMP at lower resolution to provide a framework for how the protein might work. The protein forms dimers, with two transport domains positioned at dimer interface with the S1-S4-like domains in the periphery. The apo structures are similar and show how the S1-S4 domain forms a voltage-sensing domain with all the essential features known from extensive studies on related voltage-activated cation channels, but instead of coupling with a pore forming domain the S4 helix projects outside the membrane into the intracellular regions to insert into a tunnel formed between a series of intracellular helices and the CNBD. In these apo structures, the transport domains are in an inward facing conformation similar to previously seen in many other transporters adopting the NhaA fold with conserved Na binding sites deep within the TM regions and open to the intracellular side of the membrane. The dimer interface contains a hydrophobic cavity seen in related transporters and in the nanodisc structure phosphatidic acid lipids are bound near the interface. In previous functional studies cAMP diminished the extent of hyperpolarization needed to activate the transporter, so the authors reasoned that structures with cAMP bound at 0 mV might provide information on the nature of the conformational change involved in activation of the transporter. In the presence of cAMP, density for the CNBD is not seen, but the authors discern movements within the ICHs and S4 that lead them to propose that membrane hyperpolarization induces an inward movement of S4 to diminish interactions between the intracellular domain and freeing the transport module to exchange Na and protons. In effect, the proposal is that inward movements of S4 diminish an inhibitory interaction between the cytoplasmic domain and the transport domain, thus enable the transport domain to exchange Na and protons. This mechanism is presented as an initial framework for thinking about the overall mechanism, but it is the apo structures that provide the most valuable foundation for understanding this hybrid family of cyclic nucleotide and voltage regulated transporters. Overall, the study appears to be of very high quality, the exposition lucid and well-written and the conclusions nuanced and for the most part are appropriately cautious.

Thank you!

The following are suggestions for improving the manuscript in revision.

1) To improve the resolution of VSD, have the authors tried or considered using a tight mask around the VSD to do 3D Classification without image alignment after particle expansion? This might result in higher resolution and show different conformations of the VSD.

Thank you for this very helpful suggestion. We could indeed improve the map density for the VSD when further carrying out masked refinement after symmetry expansion. We can still only observe the up configuration for S4 in the VSD, however, we find that map density supports a repositioning of the two gating charge arginine 803 and 806, such that they now more similar to the position in VSD_{IV} of Nav1.7. We have included Supplementary Movie 1 to show the model to map fitting of the VSD from the masked-refinement map after symmetry expansion.

2) It would be help for the authors to add figures highlighting the interactions between S4 and the ICH and CNBD. What kinds of residue interactions are present in this region and what might be said about how they would need to rearrange if we assume S4 moves perpendicular to the membrane by about 15-20 Å?

This is a central point and in hindsight we should have been clearer here. The extra membranous S4 region makes extensive contacts (hydrophobic, hydrogen bonds, salt-bridges) to ICH3, ICH4, ICH5, ICH6 and the CNBD. ePISA calculates that the buried surface area of the soluble S4 region (817 to 856 residues) is around 2,400 Å². To put this number into perspective the dimerization interface in the bacterial Na⁺/H⁺ exchanger NapA is only 1,900 Å². Based on these extensive interactions, we expect the ICHs will move mostly *en bloc* with S4 in response to hyperpolarization. Consistently, the cryo EM maps for the soluble portion of the S4 helix are some of the strongest features.

ePISA calculation of soluble S4 region

interface #2/9

Interface Summary XML

	Structure 1		Structure 2	
Selection range	C		A	
class	Protein		Protein	
symmetry operation	x,y,z		x,y,z	
symmetry ID	1_555		0_555	
Number of atoms				
interface	200	61.9%	270	3.4%
surface	260	80.5%	5145	64.5%
total	323	100.0%	7974	100.0%
Number of residues				
interface	40	95.2%	74	7.3%
surface	42	100.0%	987	97.8%
total	42	100.0%	1009	100.0%
Solvent-accessible area, Å²				
interface	2495.5	58.3%	2286.6	3.8%
total	4278.8	100.0%	59862.9	100.0%
Solvation energy, kcal/mol				
isolated structure	-19.0	100.0%	-909.0	100.0%
gain on complex formation	-22.4	117.7%	-19.6	2.2%
average gain	-23.0	120.5%	-20.1	2.2%
P-value	0.547		0.549	

In addition, we have further added an additional Extended Data Fig. 7 (pasted below) highlighting many of the specific interactions between the non-membranous part of S4 and the intracellular helices and the CNBD.

3) In the initial submission, it is unclear to what extent the data obtained in the presence of cAMP adds to the story given the limited resolution and missing density for the CNBDs. It would help if the authors could add a supplementary figure showing a comparison between the cryo-EM maps and models in the absence and presence of cAMP so the reader can have a better sense of what can be concluded at this stage.

Absolutely. In the absence of cAMP, 3DVA shows how the ICH7 might be coupled with the rest of the ICH network and the VSD (Supplementary Movie 2). We think the structure with cAMP helps to confirm these intrinsic motions are relevant, as well as providing new details of VSD repositioning. Furthermore, we find it fascinating that VSD starts to rotate in the same manner as the VSD in NaV channels, upon addition of cAMP.

SLC9C1 in GDN (transparent) overlaid with SLC9C1 in presence of 100 μM cAMP (colored).

We have further clarified the likely coupling between cAMP binding to the CNBD and S4 movement. The CNBD is a well-conserved fold with the residues known to coordinate cAMP conserved between the CNBD in SLC9C1 and the CNBD in ion-channels, e.g, HCN2 and MloK1 channel shown below and now included as Extended Data Fig. 8b.

It is known that the binding of cAMP causes a clam-shell closing of the α C helix (highlighted above). The main interactions between S4 and the CNBD is a salt-bridge formed between E1105 in the α C helix and Lys852 in S4. There are further polar interactions between two glutamates in S4 and the α D helix. The most probably sequence of events is that when cAMP binds, the inward movement of the α C helix disrupts the interactions between the CNBD and S4. As a consequence, the CNBD domain becomes more flexible and CNBD-CNBD interactions are also disrupted. Consistently, in the presence of cAMP, S4 has moved away, together with the ICH3, ICH4 and ICH5 helices that are interacting extensively with the non-membranous S4.

We have now included the figure above in Extended Data Fig. 10b.

Is there a reason the authors used a relatively low concentration of cAMP (0.1 mM) as opposed to the higher concentrations used in ref 5 for functional studies or from related work on HCN channels?

Since the reported cAMP K_d for CNBD is between 2 to 15 μM ²⁻⁵ and because at 100 μM we are already at a 5-fold higher molar concentration than SLC9C1, we thought that this would be a high enough concentration to observe cAMP binding. We were not expecting not to be able to model the CNBD. We have further repeated the structure of SLC9C1 with 0.1 mM cAMP into nanodiscs to 4.2 Å resolution and observe the same conformational differences as in detergent with 0.1 mM cAMP (see below). Since the functional studies were carried out with 1 mM cAMP⁶, we have now also repeated the structure of SLC9C1 in detergent with 1 mM cAMP to 3.4 Å. When contoured to the same sigma level we obtain similar map features to those in the initial submission using 0.1 mM cAMP.

0.1 mM cAMP GDN
0.1 mM cAMP nanodisc
1.0 mM cAMP GDN

We have further clarified that we see some map features for one of the CNBDs at a low map threshold for the 0.1 mM cAMP collection (the same CNBD connected with the modelled VSD'). This analysis is now shown in Extended data Fig. 10a. However, the map density was too poor to rigid body fit the CNBD, and for this reason no CNBDs were modelled.

poor map features for one of the CNBDs that also show map density for VSD'

Can the authors provide more information on how cAMP influences the biochemical properties of the SLC? Also, the SDS-PAGE gel shown in Supplementary Figure 11 reveals a band between 25 and 37 kDa, which makes one wonder whether the protein has undergone proteolytic cleavage and that is why density for the CNBD is missing.

Thank you for noticing this. We agree that this band looks like it might have been breakdown product. We have now included non-cropped gels in the Extended Data and pasted here (annotated versions will also be included with all source data). The band between 25 and 37 kDa is a likely contaminant (heat-shock protein) that is present in all purification steps. We have further clarified in the methods that cAMP was only added to the sample prior to blotting and therefore the incubation time was less than 5 mins.

We have further incubated SLC9C1 with 1 mM cAMP for 2 hours at 4°C and then run these samples on an SDS-PAGE gel. As shown below, we see no obvious degradation of SLC9C1 in detergent.

1: D238A
 2: WT without cAMP
 3: WT with cAMP

A similar band is seen in Supplementary Figure 3 but that is for the sample in nanodiscs and that band was assigned to MSP1E2. The samples in Supplementary Figure 11 are in GDN and that part of the gel is not shown for the apo protein in GDN in Supplementary Figure 2.

In the uncropped gels it becomes obvious that we remove this contaminant when the protein is incorporated into nanodiscs, and the contaminant band its migrating at a different position to the MSP1E2 protein (pasted above).

Even if this part of the manuscript can be strengthened, the authors should be quite clear that any proposed mechanism for how S4 movement controls the transporter are speculative and would require considerably more extensive efforts.

Whilst we agree the activation model is an interpretation based on the an inactive SLC9C1 structure, because of the i. high structural similarity of the VSD in SLC9C1 and the ii. available structures of many VSDs, we think it is clear that S4 in SLC9C1 will similarly move to the down-state, upon hyperpolarization. Comparing the structure with previous electrophysiology data means that the vertical displacement has to be about 10 Å for the gating charge Arg803 to traverse the transmembrane electric field; substitution of Arg803 to glutamine decreases a whole gating charge⁶. Because of the extensive interactions of the ICHs to the solvent exposed S4 region and, because there is no S4-S5 linker, these parts will move together largely *en bloc* with S4.

SLC9C1 is inactive at 0 mV. There are only two main models for inhibition. Either the ion binding site in SLC9C1 is inaccessible or the transporter is restricted from moving. In the SLC9C1 structure, the ion-binding site is accessible to the cytoplasm and SSM-based electrophysiology supports it can bind Na⁺ at 0 mV, which means the lack of activity must be due to the later possibility. That is, physical locking of the SLC9C1 transporter. The only direct interactions to the transporter module are made by ICH7, which is positioned between the ICHs, in turn interacting with the soluble S4 helix and the CNBD. 3DVA shows that in states where ICH7 is further from the transporter the map density for the SLC9C1 transporter module is poorer (Supplementary Movie 3). Taken together, we propose that when S4 moves to the down-state, ICH7 will move with it, removing this constriction.

Consistent with this activation mechanism, 3DVA of the structure with cAMP shows that ICH7 is more mobile and has a higher population of states that show detachment from the transporter (Supplementary Movie 3). We think this explains why cAMP addition enables SLC9C1 activation at less negative membrane voltages. Thus, although we fully agree that further structures and extensive mutagenesis and functional data are required to fully tease out the exact details for S4 activation of SLC9C1, we think the conceptual framework of the model shown here is strongly supported by the data currently available to us.

4) The experiments reported in Supplementary Figure 7C where the authors examine the influence of lipids on the thermal stability of the protein seem problematic. In these experiments, FSEC is used to detect the stability of protein after adding lipid and heating. However, the total amount of protein (aggregation + dimer + monomer) in each condition is not equal. For example, the sample without heating should have the biggest area, but it's much

smaller compared to that of sample in heat + DDM + DOPA, for example. I am not sure what these data say or how important they are for the study, but this needs to be better controlled and how heat might influence the fluorophore needs to be addressed.

Using purified GFP-fusion tags, we previously shown FSEC-TS can be used to monitor specific lipid interactions to membrane proteins^{1,7,8}. We have used this approach to show cardiolipin-specific stabilization of the Na⁺/H⁺ antiporter NhaA⁷ as well as other lipid interactions in SLC9B2⁹ and SLC9A9¹ Na⁺/H⁺ exchangers. We have optimized the thermal-shift assay for DDM purified proteins⁸ and, for this reason, we repeated the purification of SLC9C1 with DDM/CHS rather than GDN. As shown in the FSEC traces pasted below, when extracted from membranes by DDM/CHS the SLC9C1 protein starts to dissociate into monomers, whereas in GDN we can retain a stable homodimer. This observation was already indicated to us that specific lipids might be stabilizing the protein, i.e., respiratory protein supercomplexes dissociate if they are extracted in DDM, whereas GDN will retain stable complexes that are glued together by cardiolipin¹⁰. Although the SLC9C1 protein is less stable in the detergent DDM/CHS and the FSEC traces are messier than those with GDN, we can better evaluate the propensity of different lipids to help stabilize the protein.

In these experiments below DDM/CHS purified SLC9C1 is heated at 45°C for 10 mins (GFP retains its fluorescence until 76°C) and then any heavy aggregates (precipitation) are removed prior to injection on the SEC column. As such, the total fluorescence signal may not necessarily be the same in all samples, since denser membrane-protein GFP aggregates may have been removed by centrifugation.

However, we think the confusion here comes from the fact that SLC9C1 appears to be more stable at 45°C with DOPA than it does at 4°C without DOPA present.

In fact, this is the case. In hindsight we should have included FSEC traces of SLC9C1 with DOPA at 4°C. We have repeated these experiments. As shown below, just the addition of DOPA to SLC9C1 (but not DDM or POPC addition) stabilizes the protein at 4°C. Thus, SLC9C1 is more stable at 45°C with DOPA than it is at 4°C without DOPA, but it is less stable at 45°C than at 4°C with DOPA.

We have quantified the average peak height difference of the SLC9C1 homodimer under these different conditions from several different purifications. Overall, the addition of DOPA consistently stabilizes the SLC9C1 homodimer in detergent.

5) The supported bilayer experiments are a reasonable addition, but as the authors acknowledge they only demonstrate that the protein is capable of binding Na ions in a pH dependent fashion. The authors state that these experiments do not demonstrate that the expressed and purified protein is capable of transport, but they could also remind the reader that the protein construct they used for structure determination has been shown to be fully functional when expressed in mammalian cells (ref 5).

Thank you, that's a good point. We have clarified that recombinant *sea urchin* SLC9C1 is functional in mammalian cells. We have also included the FSEC traces of GDN-solubilized SLC9C1 WT and the D238A variant to show that they are both well-folded upon detergent extraction from membranes.

6) The structure of the voltage-sensing domain in SLC9C1 is most similar to that found in Nav1.7, so it makes sense that the Nav channel is compared most extensively, but most of the foundational work on voltage sensing comes from work on Shaker Kv channels and in places it might be worth making that connection. A structure of

Shaker is now available and the best estimates of charge per channel were done on that protein (PMIDs 8663993 and 8663992), for example.

Good idea. We now introduce the VSD domain in SLC9C1 as harboring the prototypical Shaker Kv VSD (r.m.s.d. of 2.5 Å to PDB ID: 7SIP) and the superimposition is included in the Extended Data Fig. 6 next to the sequence alignment of the VSDs that includes the *drosophila* Shaker Kv sequence.

7) Lines 131 and 132 could be more clearly explained and I assume the amount of protein in these experiments is fixed?

When altering the lipid-to-protein ratio for SSM-based electrophysiology experiments the total amount of lipid is fixed, but the amount of protein added decreases. If there is no transported charge then the peak signals should be linear with the amount of protein added. On the other hand, if there is a transported charge, then the build-up of a membrane potential will slow down charge accumulation and the slope of the decay currents (re-equilibration) will change. When comparing normalized peak currents, we only observe no systematic change in the slope of the decay currents at different LPRs, which is most consistent with binding rather than transport. LPR 500 shows a small difference in decay currents, however, as the peak currents for LPR 500 are indistinguishable from the D238A variant at LPR5 we think the signals are probably too low to accurately compare and have not included them in this analysis. We now include the decay time constants from averaged titrations at each LPR to show they are within error from one another (the decay times of the transient currents are derived from monoexponential fits of the current decay).

We have included more details in the main text, updated Supplementary Fig. 2 and Supplementary Fig. 3, figure legend and methods.

8) Line 289 ICH3-ICH3 should be ICH3-ICH4, correct?

Yes, thank you for spotting this error.

It looks like the intracellular extension of the S4 helix 'tunnels' into the intracellular domain between the ICH helices and the CNBD. The authors might consider using that description if it is indeed correct.

Nice suggestion we have used the work tunnels (line 282)

9) Line 329 typo "semes"

Thanks, typo fixed.

10) The authors often use words like above or below to orient readers when it would probably be easier for the general audience if the authors used terms like internal/external or intracellular/extracellular to describe relative relationships.

11) Pg 8 line 210 at the end I think "other" is missing after "each".

Thanks, typo fixed.

12) CNB domain and CNBD are used interchangeably, and it would be good to stick with one, for example by first defining CNBD and then using that nomenclature throughout.

Thank you, we went with CNBD to keep the word count down.

13) some of the blue colors in Fig 1 are a bit challenging to distinguish and other choices might make it easier to comprehend.

Thank you. These colors have been changed.

Referee #2 (Remarks to the Author):

The manuscript by Yeo et al presents several novel structures of a sperm-specific sea urchin hyperpolarization-activated and cAMP-modulated Na/H antiporter that contains, in addition to the transporter core, both voltage-sensor and cyclic nucleotide-binding domains. The structures are solved in both detergent and nanodiscs in the presence and absence of cAMP, they are of high resolution, and they support an exciting novel mechanism for how cAMP and voltage work together to modulate the function of transporters in this family. The S4 transmembrane helix of the voltage sensor domain, which is of extended length compared to S4 helices in other channels (~60 Å beyond the membrane) is clamped between two sets of helix-turn-helix structures that are part of an extended network of helices that forms a tight dimer interface beneath the transporter domain dimer. These helices connect to the cyclic nucleotide-binding domains and additional structures that further contribute to the dimer interface. The authors propose that at 0 mV, the helical network is sterically blocking the core domain conformational changes required for transport (ICH7 blocks TM7 movement). Addition of cAMP leads to increased dynamics in the CNBDs (so much so that the density for CNBDs is no longer defined in the cAMP-bound structure), perturbs the dimer interface and the helical network (that obstructed transporter motions) enough so that when a hypothetical downward movement of the long S4 upon hyperpolarization occurs, the inhibition of the transporter by the helical network would be removed and the transporter becomes active. The authors perform a thorough comparative analysis of this novel quirky transporter with other transporters in the family, and with voltage-gated channels (for the voltage sensor) and CNBD-containing channels (for the CNBD). The structures are impressive, the manuscript is well-written and presented, the figures are beautiful. I have only a few minor points, listed below, and a concern regarding the functional assays.

Thank you!

Concerns:

To assess the functionality of their purified transporter, the authors perform SSM-based electrophysiology and they state that their current measurements likely measure the binding of Na to the transporter and not transport activity, which the authors claim it is supported by increase in peak current with increase in Na concentration and no change in the time constant of decay when they decrease the lipid-to-protein ratio (LPR) (Fig.1). However, neither Fig. 1d, e, or c show that Na binding is mediated by the transporter, which is concerning. Non-specific binding to the bilayer is consistent with the peak current going up with Na concentration. The lower signal from the D238A mutant in only one Na concentration is not very compelling because very little detail is given of this mutant (sample concentration and quality, LPR, function, etc) and there is only one Na concentration tested.

Thank you for your feedback. To Figure 1c, we now include the traces for empty liposomes, which shows that the Na⁺-induced currents are pH dependent for SLC9C1, whereas the empty liposomes show less pH dependence.

For Figure 1d, we now include in the Supplementary Fig. 3. the raw peak currents from (one out of the two technical repeats) made on each of the 3 different sensors that was used to generate the binding curves shown in Figure 1e

Suppl Fig. 3

SSM-based electrophysiology measures capacitance coupling. It is possible at higher Na⁺ concentrations causes a leakage of the liposomes and we observe this effect as non-protein mediated charge accumulation. In transport assays using isotope-labelled substrates we also often observe non-protein-mediated substrate accumulation in empty liposomes whilst assaying sugar GLUT transporters, which similarly have weak affinity for their substrates (in the mM range).

The differences between the peak currents from SLC9C1 WT and the D238A mutant, however, show that the peak currents are consistently much higher for the WT protein. To provide further support for this conclusion, we have now included the SSM-based electrophysiology responses to the related Na⁺/H⁺ exchanger NHE9 with the ion-binding site aspartic acid substituted to alanine (non-functional mutation) and the unrelated sugar transporter GLUT5. The responses at different Na⁺ concentrations match the peak currents observed from either the D238A mutant or empty liposomes. The titrations have been made on multiple sensors to account for background variation. Because there are so many titrations, we prefer to show the average titration curves.

The combined titrations from individual sensors. We see clearer Na⁺ saturation with the lower LPR 125 than LPR 5 (shown below) and have therefore only included the Na⁺ binding data for LPR 125 in the manuscript.

As suggested by referee 1, we now also mention in the text the *sea urchin* SLC9C1 is active when recombinantly expressed in mammalian cells in line 110.

Fig. 1c inset shows only a normalized signal and the authors should definitely show the peak currents here, since the signal should change predictably with changes in LPR. Furthermore, the authors should state the number of biological repeats performed here (number of protein preparations tested).

Your'e right, we should have also included the non-normalised peak traces. These have now been added and included here in response to referee 1, Q7.

Lines 320-323: Just because the CNBs cannot be modelled, one should not assume that the entire interface disappears, as the authors do. At the very least, an exact reduction in the value of buried surface area should not be stated.

In hindsight, we see your point. We meant that the CMBDs were no longer contributing to stabilizing the homodimer. We have removed the ePISA comparison to avoid any confusion of this point.

I did not see a good comparison between the CNBD of the SLC9 and that of other CNBD-containing ion channels such as HCN, in various activated, resting, etc states. Overlay (better than that in Supplementary Fig. 10) would be useful.

Thank you for this suggestion. Please see responses to referee 1.

Supplementary Fig 4a labels a PC lipid in the GDN structure. However, there is no other supporting evidence suggesting that the lipid is a PC. Unless the authors have further evidence, I recommend removing the label. They can speculate it is a PC in the text, but not label it as such.

Fair point. We have clarified the main text that we have modelled a putative PC based on the fact the we see head-group density and this is the most abundant lipid. However, the region lacks charged residues and so we agree we cannot rule out that is a different lipid than PC.

Referee #3 (Remarks to the Author):

Summary

The manuscript by Yeo and colleagues describes the first high-resolution cryo-EM structure of the sperm-specific voltage-gated Na⁺/H⁺ exchanger SLC9C1, using the sea urchin gene as source material. The structure of this transporter is of considerable interest physiologically and to the broader pharmaceutical industry as loss-of-function of SLC9C1 in mammals impairs sperm motility and fertilization and thus a prime target for the development of male contraceptives.

Main Comments:

1. This study is original and the manuscript is well written. The experimental design is sound, the data are compelling and the conclusions are generally sound, with some reservations described below (in Other Comments). While structures for other Na⁺/H⁺ exchangers from prokaryotes and eukaryotes, including mammals, have been reported previously, what makes the structure of this transporter unique is the presence of a membrane voltage-sensing domain (VSD) and a cytoplasmic cyclic nucleotide (cAMP) binding domain (CNBD), features unparalleled amongst Na⁺/H⁺ exchangers as well as other secondary active transporters. Based on the structure, the authors provide significant biological insight into the mechanics of the VSD – a domain that shares homology to the VSD of the human voltage-gated Na⁺ channel Nav1.7 - as well as partial insight into the dynamics and role of cAMP binding to the CNBD and their impact on Na⁺/H⁺ exchange. Overall, this is a very meritorious study that advances our understanding of this structurally diverse family of cation/proton exchangers and should be of broad interest to the scientific community and society in general.

Thank you!

Listed below are other comments that require attention, clarification and/or experimentation.

Other Comments

1. Sea Urchin should be included in the title to indicate the biological source of the structure.

We thought along the same lines, but we felt that the title gets too long. We specify in the abstract the organism is from Sea urchin. While we would prefer to keep the title as we have written it, we are not strongly against changing it if you think this is important to clarify the organism in the title too.

2. Please clarify the statement on lines 125-126 "In contrast to SLC9A and SLC9B members that display maximal activity at pH 8.5..." The basis for this statement is unclear as the activities of all SLC9A and SLC9B members assessed to date are driven by the relative concentration gradients of Na⁺ and H⁺, and thus are more active at high [Na⁺] and acidic pH rather than alkaline pH. I could find no information in the cited review (Ref. 11) indicating that SLC9A and SLC9B members are maximally active at alkaline pH 8.5.

Thank you for spotting this error. We should have written pH 7.5 and not pH 8.5, but we may have also further over-simplified this statement.

As you know, because you need to induce acid-loading in cells to stimulate NHE exchange activity, then it is difficult to estimate the pH dependence of NHE activity in cells using traditional approaches. By using ion-selective electrodes and whole-cell patches with variable internal (and external) pH, you can, however, clamp the conditions on both sides of the membrane and you can study the kinetics in a more controlled system. With this setup, NHE1 has a maximal activity with 140 mM external NaCl at pH 7.6. In the forward transport mode, the half-maximal extracellular Na⁺ concentration shifts from 78 mM at a cytoplasmic pH of 6.0 to 34 mM at a cytoplasmic pH of 7.6¹¹.

The transient currents measured in these experiments by Daniel Fuster are consistent with the activity measured for NhaA and NapA bacterial Na⁺/H⁺ exchangers in proteoliposomes, which show increased Na⁺ transport between pH 6.5 and pH 7.5, e.g.,¹²⁻¹⁴. We have further measured transient currents of SLC9B2 (NHA2) proteoliposomes by SSM-based electrophysiology with increasing activity from pH 6.5 to pH 7.5 (pasted below from ref⁹).

Taken together, our main point was to highlight that SLC9C1 shows better Na⁺ binding at pH 6.5 rather than pH 7.5, which shows higher activity for homologues proteins measured under similar conditions. We have re-written this sentence accordingly.

3. In Figure 1d, why does the decay of the peak current at the different [Na⁺] and pH 6.5 not return to baseline in contrast to data shown in Fig. 1c?

For the experiments in Figure 1c, we used 3 mm sensors, whereas in Figure 1d we used 1 mm sensors. Because the signal-to-noise ratio in the 1 mm sensors was better than the 3 mm sensors, we decided to use the 1 mm sensors in most experiments. To be more consistent in Figure 1c, however, we now show the LPR screening using 3 mm sensors (see response to referee 2).

During the course of our project Nanion changed the surface coating on the 1 mm sensors. With the new formulation the baseline returns to zero now with the 1 mm sensors as it does with the 3 mm sensors (both with

SLC9C1 and empty liposomes). Since we are only analysing the peak height, this doesn't affect our analysis of the Na⁺-induced currents.

4. In the text (lines 134-135 and 207-209) and Figure 1e, the authors indicate that sea urchin SLC9C1 Asp238 (TM6) is a strictly conserved aspartate residue in Na⁺/H⁺ exchangers that is essential for ion-binding and transport. Indeed, an Asp residue is conserved at the equivalent position in all members of the SLC9A family, and the authors back this assertion by showing that a D238A mutation in sea urchin SLC9C1 negates Na⁺ dependent currents (Figure 1e). However, what is perplexing is that D238 is not conserved in SLC9C1 from other species, such as humans and platypus which possess a Ser (Ser179) and Ala (Ala230), respectively, at the equivalent position (see sequence alignments in supplementary Fig. 1.). How can Asp be critical if Ser and Ala are tolerated in other species. Please comment.

We contemplated discussing this point and we are pleased that you brought this up as we also find it interesting. Across the large family of Na⁺/H⁺ exchanger the ion-binding aspartate is (almost) strictly conserved¹⁵. For NHEs and related bacterial Na⁺/H⁺ antiporters harboring the ion-binding aspartate, then the aspartate residue has shown to be essential for ion binding and transport. Interestingly, the ND motif^{15,16} is not strictly conserved in the mammalian SLC9C1 members and is replaced in human SLC9C1 by "TS" (See alignment pasted below).

However, in the mammalian SLC9C1 homologues there is now an additional glutamate in the cross-over helices that is not present in other SLC9C members or, in fact, any other SLC9A-SLC9B members that I am aware of. This would provide the acidic required coordinating Na⁺. It is unclear the reason for this divergence. It's an interesting difference that we will be exploring further, but so far, have not been able to express the mammalian SLC9C1 members to tackle this question.

We have added this information to the Supplementary Fig. 1 legend. The main point in the main text was to emphasize that the ion-binding site was very similar to that seen in other NHE1 and NHE9 structures and therefore should coordinate Na⁺ in a similar manner.

In regard to the above, the authors should provide some western blot data showing the protein expression of D238A, and evidence that the equivalent amounts were incorporated into the proteoliposomes for functional measurements.

The SLC9C1 WT and D238A variant are GFP-tagged. As shown below, we extract roughly the same amount of the SLC9C1-GFP homodimer from membranes using the detergent GDN. As shown in the gel below for LPR 5 we find that the reconstitution efficiency is about 30%. We do not observe any clear differences between the reconstitution efficiency of SLC9C1 WT and the SLC9C1 D238A variant. In addition, we have also included the SSM-based electrophysiology responses to Na⁺ for the ion-binding aspartate mutant in NHE9 (that we previously showed was non-functional¹) as well as an unrelated sugar transporter GLUT5 (see responses to referee 2)

The quality of WT and D238A variant is similar as shown here after re-injection of the final SEC peak collected for functional analysis.

5. Lines 325-328: The authors refer to a highly-conserved proline-rich linker (949-PPPPP-953) that connects ICH7 with the CNBD's, yet according to the alignment in supplementary Fig. 1, this sequence appears to be present only in sea urchin SLC9C1, but not SLC9C1 from other species. Can this really be referred to as "highly-conserved" and hence is it biologically significant?

Apologies for not being clear here. This exact proline-stretch is only present in *Sea urchin* (at least from the sequences shown here), but many organisms have three proline residues along this region, which is fairly unusual and what we meant with highly-conserved proline-rich linker.

		ICH7										**	**	αD	*																						
sea urchin	926	E	A	H	K	L	E	L	T	V	E	I	K	M	K	R	L	M	N	A	P	S	S	I	P	P	P	P	P	E	N	L	L	K	N	V	S
human	835	E	G	A	G	I	N	K	L	I	M	A	K	K	K	E	V	L	D	S	Q	S	I	I	R	P	L	T	V	E	E	V	L	Y	H	I	P
cow	841	E	G	S	E	I	N	K	I	I	M	A	K	K	R	D	I	L	D	F	Q	P	T	Y	K	P	L	T	V	E	E	A	L	Y	H	I	P
rat	855	E	G	A	E	I	N	K	L	I	M	A	K	K	K	Q	V	L	E	L	Q	S	V	I	Q	P	L	N	V	E	E	A	P	Y	Y	I	P
platypus	866	E	S	S	G	M	H	K	M	I	L	F	K	R	K	A	M	L	D	F	P	S	F	I	K	A	P	S	I	K	E	M	L	N	H	V	T
orcha	858	E	G	F	E	I	N	K	L	I	M	A	R	K	R	E	I	L	D	L	Q	P	M	Y	K	P	P	T	V	D	K	A	L	Y	H	I	P
bamboo shark	877	E	A	L	K	L	E	T	M	I	E	V	K	M	K	R	L	L	K	F	P	P	S	I	Q	P	P	T	A	E	E	L	L	K	N	L	P
pond turtle	860	E	E	A	K	L	Q	K	M	I	L	Q	K	K	H	N	L	G	T	L	P	S	T	I	A	P	P	T	A	E	E	L	L	H	S	I	S
alligator	852	E	G	A	K	I	E	K	M	I	L	I	K	K	K	Q	L	G	T	L	P	S	T	I	A	P	P	T	A	E	E	L	L	R	N	I	S

Noticeable, however human SLC9C1 has only 1 conserved proline residue. To more accurately reflect the sequence conservation, we have modified the sentence accordingly.

6. Line 956: states that R399 is in TM12, but in the supplementary Fig. 1 alignment, R399 is located in TM11.

Yes! Thank you for spotting this typo.

7. In the legend to supplementary Fig. 1, please define the acronym labels, e.g. ECH1, ICH1-7 etc...

Thank you. This information has now been added.

John Orlowski

- 1 Winkleman, I. *et al.* Structure and elevator mechanism of the mammalian sodium/proton exchanger NHE9. *The EMBO journal* **39**, e105908 (2020). <https://doi.org:10.15252/emboj.2020105908>
 - 2 Kusch, J. *et al.* Interdependence of receptor activation and ligand binding in HCN2 pacemaker channels. *Neuron* **67**, 75-85 (2010). <https://doi.org:10.1016/j.neuron.2010.05.022>
 - 3 Lolicato, M. *et al.* Tetramerization dynamics of C-terminal domain underlies isoform-specific cAMP gating in hyperpolarization-activated cyclic nucleotide-gated channels. *J Biol Chem* **286**, 44811-44820 (2011). <https://doi.org:10.1074/jbc.M111.297606>
 - 4 Porro, A. *et al.* The HCN domain couples voltage gating and cAMP response in hyperpolarization-activated cyclic nucleotide-gated channels. *eLife* **8** (2019). <https://doi.org:10.7554/eLife.49672>
 - 5 Lorenz, R. *et al.* Mutations of PKA cyclic nucleotide-binding domains reveal novel aspects of cyclic nucleotide selectivity. *The Biochemical journal* **474**, 2389-2403 (2017). <https://doi.org:10.1042/BCJ20160969>
 - 6 Windler, F. *et al.* The solute carrier SLC9C1 is a Na(+)/H(+)-exchanger gated by an S4-type voltage-sensor and cyclic-nucleotide binding. *Nat Commun* **9**, 2809 (2018). <https://doi.org:10.1038/s41467-018-05253-x>
 - 7 Nji, E., Chatzikyriakidou, Y., Landreh, M. & Drew, D. An engineered thermal-shift screen reveals specific lipid preferences of eukaryotic and prokaryotic membrane proteins. *Nat Commun* **9**, 4253 (2018). <https://doi.org:10.1038/s41467-018-06702-3>
 - 8 Chatzikyriakidou, Y., Ahn, D. H., Nji, E. & Drew, D. The GFP thermal shift assay for screening ligand and lipid interactions to solute carrier transporters. *Nat Protoc* **16**, 5357-5376 (2021). <https://doi.org:10.1038/s41596-021-00619-w>
 - 9 Matsuoka, R. *et al.* Structure, mechanism and lipid-mediated remodeling of the mammalian Na(+)/H(+) exchanger NHA2. *Nat Struct Mol Biol* **29**, 108-120 (2022). <https://doi.org:10.1038/s41594-022-00738-2>
 - 10 Brzezinski, P., Moe, A. & Adelman, P. Structure and Mechanism of Respiratory III-IV Supercomplexes in Bioenergetic Membranes. *Chem Rev* **121**, 9644-9673 (2021). <https://doi.org:10.1021/acs.chemrev.1c00140>
 - 11 Fuster, D., Moe, O. W. & Hilgemann, D. W. Steady-state function of the ubiquitous mammalian Na/H exchanger (NHE1) in relation to dimer coupling models with 2Na/2H stoichiometry. *The Journal of general physiology* **132**, 465-480 (2008). <https://doi.org:10.1085/jgp.200810016>
 - 12 Winkelmann, I. *et al.* Crystal structure of the Na(+)/H(+) antiporter NhaA at active pH reveals the mechanistic basis for pH sensing. *Nat Commun* **13**, 6383 (2022). <https://doi.org:10.1038/s41467-022-34120-z>
 - 13 Uzdavinyis, P. *et al.* Dissecting the proton transport pathway in electrogenic Na(+)/H(+) antiporters. *Proceedings of the National Academy of Sciences of the United States of America* **114**, E1101-E1110 (2017). <https://doi.org:10.1073/pnas.1614521114>
 - 14 Mager, T., Rimon, A., Padan, E. & Fendler, K. Transport mechanism and pH regulation of the Na+/H+ antiporter NhaA from Escherichia coli: an electrophysiological study. *J Biol Chem* **286**, 23570-23581 (2011). <https://doi.org:10.1074/jbc.M111.230235>
 - 15 Masrati, G. *et al.* Broad phylogenetic analysis of cation/proton antiporters reveals transport determinants. *Nat Commun* **9**, 4205 (2018). <https://doi.org:10.1038/s41467-018-06770-5>
 - 16 Brett, C. L., Donowitz, M. & Rao, R. Evolutionary origins of eukaryotic sodium/proton exchangers. *Am J Physiol Cell Physiol* **288**, C223-239 (2005). <https://doi.org:288/2/C223> [pii]
- 10.1152/ajpcell.00360.2004

Reviewer Reports on the First Revision:

Referees' comments:

Referee #1 (Remarks to the Author):

The authors have done a really nice job of revising the manuscript to address the points I made in my original review and I have no further suggestions. The authors are to be congratulated on a really fascinating story, and its nice to see such thoughtful transporter aficionados entering the voltage-sensing community!

Referee #2 (Remarks to the Author):

The authors have addressed all my questions and concerns. I highly recommend this paper for publication in Nature.

Referee #3 (Remarks to the Author):

The revised manuscript by Yeo and colleagues have provided very satisfactory and detailed responses, including additional data, to all concerns raised in the initial review. The statistical analyses are appropriate and clearly documented. Overall, this is a very meritorious study that should be of broad interest to the scientific community.